# Combining LOPIT with differential ultracentrifugation for high-resolution spatial proteomics

Aikaterini Geladaki [1,2], Nina Kočevar Britovšek[1], Lisa M. Breckels[1], Tom S. Smith [1], Owen L. Vennard [1], Claire M. Mulvey[1], Oliver M. Crook [1,3], Laurent Gatto [1,4] & Kathryn S. Lilley [1]

The study of protein localisation has greatly benefited from high-throughput methods utilising cellular fractionation and proteomic profiling. Hyperplexed Localisation of Organelle Proteins by Isotope Tagging (hyperLOPIT) is a well-established method in this area. It achieves high-resolution separation of organelles and subcellular compartments but is relatively time- and resource-intensive. As a simpler alternative, we here develop Localisation of Organelle Proteins by Isotope Tagging after Differential ultraCentrifugation (LOPIT-DC) and compare this method to the density gradient-based hyperLOPIT approach. We confirm that high-resolution maps can be obtained using differential centrifugation down to the suborganellar and protein complex level. HyperLOPIT and LOPIT-DC yield highly similar results, facilitating the identification of isoform-specific localisations and high-confidence localisation assignment for proteins in suborganellar structures, protein complexes and signalling pathways. By combining both approaches, we present a comprehensive high-resolution dataset of human protein localisations and deliver a flexible set of protocols for subcellular proteomics.

[1] Cambridge Centre for Proteomics, Department of Biochemistry, University of Cambridge, 80 Tennis Court Road, Cambridge CB2 1GA, UK. [2] Department of Genetics, University of Cambridge, 20 Downing Place, Cambridge CB2 3EJ, UK. [3] MRC Biostatistics Unit, Cambridge Institute for Public Health, Forvie Site, Robinson Way, Cambridge CB2 0SR, UK. [4] Present address: de Duve Institute, UC Louvain, Avenue Hippocrate 75, Brussels 1200, Belgium. These authors contributed equally: Aikaterini Geladaki, Nina Kočevar Britovšek. Correspondence and requests for materials should be addressed to K.S.L. (email: k.s.lilley@bioc.cam.ac.uk)

The level of complexity of the human proteome extends far beyond the number of gene products expressed by the genome in a cell[1]. The compartmentalisation within eukaryotic cells and the dynamic distribution of proteins between organelles are crucial in the regulation of cellular processes[2]. Studies of protein localisation have helped define new models to link mutations to certain disorders[3–13] and perturbations in protein subcellular localisation, in combination with abnormal expression, have been associated with many human diseases[14–19]. Thus, comprehensive subcellular maps for tissue types or cell lines under various physiological or pathological conditions have the potential to further our understanding of disease aetiology and significantly benefit drug discovery programs.

Over a decade of advances in spatial proteomics technologies has enabled the study of organelle composition, dynamics and function across a range of species and cell types[2,20]. These methods mostly rely upon centrifugation-based cell fractionation coupled with mass spectrometry (MS)-based proteomics and have been applied to characterise all major organelles, macromolecular structures and multiprotein complexes in eukaryotic cells[1,20–27]. Methods for subcellular fractionation which do not involve centrifugation have also been developed[1,26–28]. Furthermore, advances in quantitative proteomics strategies have been particularly central to the evolution of subcellular proteomics studies. In vitro stable isotope-labelling methods such as isobaric tagging are now available, allowing for the simultaneous analysis of up to 11 samples in the same experiment, and have been coupled with improvements in the accuracy of MS data acquisition[29–31]. This has enabled simultaneous quantification of a greater number of fractions per experiment, in turn avoiding unwanted technical variability between fractions analysed in separate MS runs and alleviating the issue of missing values resulting from the stochastic processes of peptide quantification by MS[29]. Major developments in bioinformatics including approaches to interrogate spatial proteomics data[32,33] and achieve sequence-based or annotation-based prediction of protein subcellular localisation[34,35] have also contributed to the evolution of spatial proteomics methods. The experimental data arising from these developments have been used to generate publicly-accessible organelle databases and web-based resources, some of which link subcellular proteomics data to functional datasets as well as disease relevance and animal model information[36–39].

Localisation of Organelle Proteins by Isotope Tagging (LOPIT) is a well-established method for the simultaneous analysis of multiple subcellular structures from complex biological mixtures in a single experiment. This contrasts with proximity tagging methods[40] which are designed to identify proteins associated with discrete cellular compartments and therefore provide protein subcellular distribution snapshots which are not easily integrated to examine proteins with multiple localisations. LOPIT does not require absolute organelle purification and is instead based on the measurement of protein distribution across multiple density gradient fractions[41,42]. In this case, subcellular localisation is assigned by comparing protein profiles to those of well-curated organelle markers using multivariate statistical analysis and machine learning approaches[33]. LOPIT has been applied to the study of the subcellular proteomes of the HEK293 human kidney cell line, DT40 chicken lymphocyte cell line, *A. thaliana* roots and root-derived callus, *D. melanogaster* embryos and *S. cerevisiae* cells[43–49]. Recently, an improved version of this method called hyperplexed LOPIT (hyperLOPIT) was developed, integrating novel approaches for sample preparation, MS data acquisition and protein localisation classification to create a high-resolution map of protein subcellular localisation in E14TG2a mouse embryonic stem cells[50].

Variations of hyperLOPIT have recently been employed by Beltran et al.[51], who integrated a temporal component to the workflow to analyse human lung fibroblast cytomegalovirus infection and Jadot et al.[52], who used Nycodenz and sucrose density gradient centrifugation to determine the rat liver organelle proteome. A label-free alternative to LOPIT, Protein Correlation Profiling (PCP), has also been developed and applied to the study of the centrosome[53] and lipid droplets[54] as well as global organelle analyses[55,56]. Additionally, PCP has been used to study the proteasome complexes of *P. falciparum*[57] and combined with Stable Isotope Labelling with Amino acids in cell Culture (SILAC) to investigate protein-protein interactions with temporal and stoichiometric resolution[58,59].

The Dynamic Organellar Maps (DOM) workflow is based on a simpler fractionation method, differential ultracentrifugation (DC), and involves separation of crude nucleus-, organelle- and cytosol-enriched fractions from SILAC-heavy cells which are then combined with SILAC-light membrane-enriched fractions[60]. DOM has recently been updated to include a TMT-labelling option in which no reference organellar sample is analysed and only the five post-nuclear fractions are used, enabling analysis of two different conditions using a single TMT 10-plex set[61]. However, this comes at the cost of reduced resolution since protein profiles are restricted to just these five fractions. A six-fraction label-free quantification approach partially rectifies this by including an additional nucleus-enriched fraction in the analysis[61]. Overall, because of the reduced number of fractions taken, this approach achieves lower resolution compared to hyperLOPIT[62].

The hyperLOPIT workflow is relatively time-consuming and requires a considerable amount of starting material. We reasoned that coupling the advances in sample preparation, MS data acquisition and protein localisation classification employed in hyperLOPIT with differential centrifugation should yield high-resolution maps of protein localisation with less starting material and at lower cost than those required by hyperLOPIT. We thus introduce Localisation of Organelle Proteins by Isotope Tagging after Differential ultraCentrifugation (LOPIT-DC), a spatial proteomics pipeline based on differential centrifugation. Crucially, unlike previous DC-based methods, LOPIT-DC enables all cellular compartments to be analysed simultaneously. We compare protein subcellular localisation maps produced by LOPIT-DC and hyperLOPIT using the human osteosarcoma U-2 OS cell line and evaluate the impact of employing a differential centrifugation-based workflow on global, experiment-wide resolution. Integrating the two approaches, we present the most comprehensive MS-based spatial proteomics map of a human cell line published to date including isoform-specific subcellular niches and localisations for large protein complexes and proteins involved in cancer-relevant signalling pathways. We further demonstrate that many proteins cannot be classified to a single localisation as they either transit between compartments or carry out their functional role(s) in multiple locations.

## Results

**Development of the LOPIT-DC method.** Aiming to create a simpler but comprehensive alternative to hyperLOPIT, we developed a second MS-based technique for the study of protein subcellular localisation which we named LOPIT-DC. To create this method, we combined the strengths of the hyperLOPIT protocol (improved cell lysis, MS data acquisition and protein localisation classification)[50,63] with differential centrifugation as employed by other subcellular fractionation workflows[52,60,61] (Fig. 1).

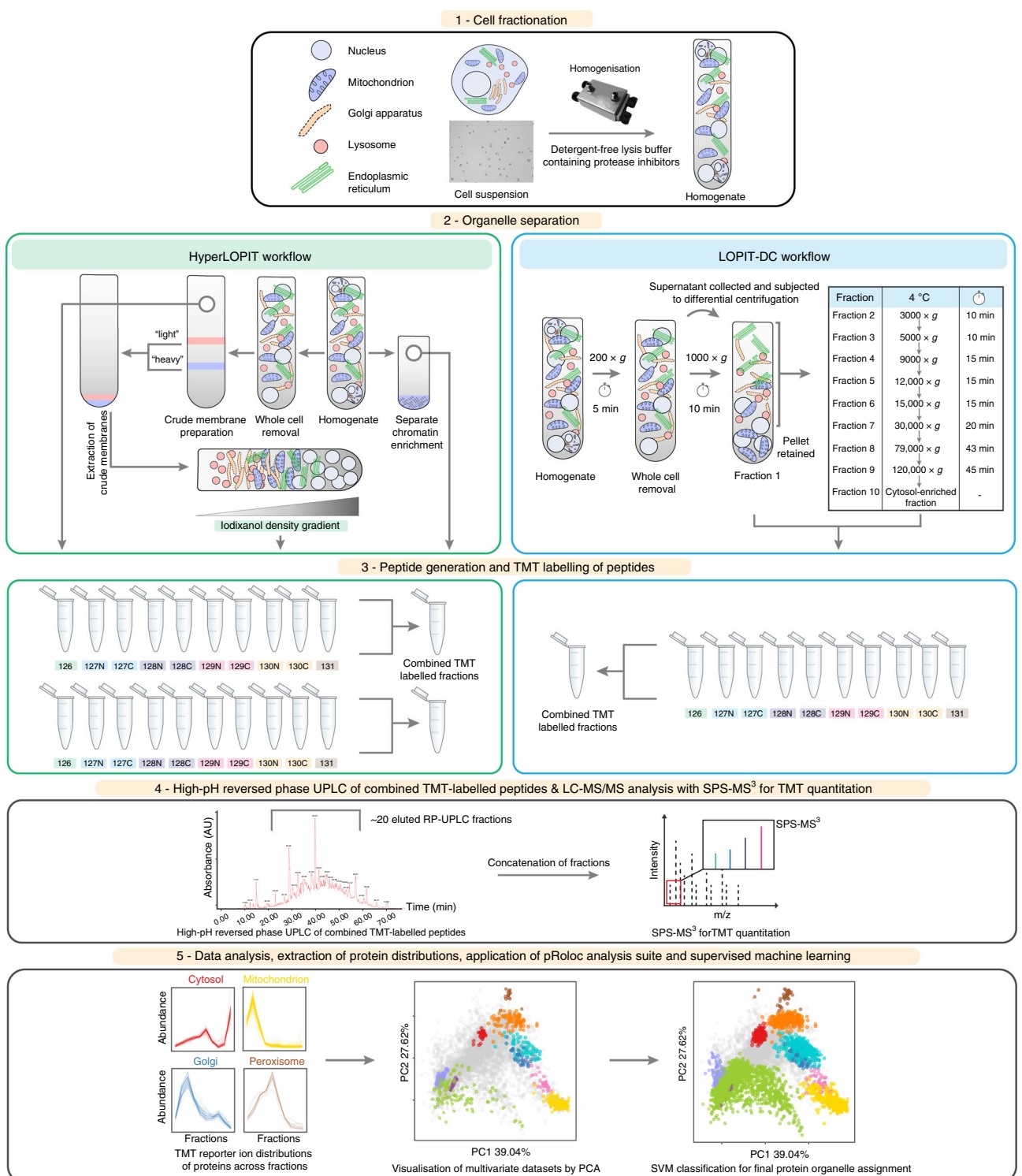

**Fig. 1** Overview of the hyperLOPIT (left) and LOPIT-DC (right) workflows. LOPIT is a quantitative mass spectrometry-based method used for the separation of organelles and other subcellular compartments. The workflows differ in step 2 (organelle separation): hyperLOPIT is based on equilibrium density gradient ultracentrifugation from a crude membrane preparation, while LOPIT-DC utilises differential ultracentrifugation following removal of unlysed cells. In addition, hyperLOPIT includes a separate chromatin enrichment step. Both workflows take advantage of multiplex TMT-labelling (step 3) to reduce mass spectrometry analysis time and technical variability and SPS-MS[3] for accurate quantification (step 4). Data analysis for both workflows is performed using pRoloc (step 5)

As in hyperLOPIT, cell lysis in LOPIT-DC is followed by a whole cell-preclearing step necessary to remove unlysed cells that could confound downstream analysis. The cell lysis stage is critical in both methods, as insufficient lysis can result in subcellular fractions with reduced protein yields while excessive lysis can damage sensitive membranes leading to organellar content release. In contrast to hyperLOPIT, where subcellular fractionation is based on density gradient ultracentrifugation, LOPIT-DC utilises sequential differential centrifugation steps to partition the cell lysate into 10 fractions. By changing the fractionation method and

excluding the separate chromatin preparation present in hyper-LOPIT, the time taken to fractionate the cells is reduced approximately three-fold in LOPIT-DC (Supplementary Table 1). Some of the centrifugation speeds in our LOPIT-DC workflow are similar to those used in DOM[60] with additional steps included to efficiently utilise the full set of 10-plex TMT and increase subcellular resolution, including an initial step to remove unlysed cells (Supplementary Table 2, Fig. 1).

Similar to hyperLOPIT, LOPIT-DC makes use of the TMT multiplexing strategy that enables simultaneous analysis of all subcellular fractions in a single MS run, avoiding technical variation and reducing missing values introduced by analysing fractions in separate MS runs. This is extremely important, as presence of many missing values in such datasets can have a detrimental effect on protein localisation assignment as recently observed by Beltran et al.[51]. Moreover, TMT-based multiplexing reduces MS run time and thus experiment cost. As with hyperLOPIT, MS analysis can be carried out using tandem mass spectrometry (MS/MS) or the SPS (Synchronous Precursor Selection)-based MS[3] technology which improves quantitative accuracy and spatial resolution[31,50]. In contrast to DOM, LOPIT-DC is an all-in-one method meaning that all subcellular niches are analysed in a single preparation.

For downstream statistical analysis, LOPIT-DC employs the same robust spatial proteomics data analysis pipeline as used in hyperLOPIT for data processing and visualisation as well as protein localisation classification[64]. These open-source, open-development R packages are interactive and new computational methods are continuously being integrated to take advantage of developments in machine learning[65].

**Application of LOPIT-DC to the U-2 OS cell line**. LOPIT-DC was applied to the human U-2 OS cell line, a well-characterised model with many reference resources. Pertinently, the Cell Atlas database includes immunofluorescence-based protein subcellular localisation data for this cell line, providing an ideal data source for orthogonal validation[38].

Three LOPIT-DC replicates were obtained, using on average $7 \times 10^7$ cells per experiment (Supplementary Figs. 1a, 2a, 3). Each replicate yielded 10 fractions with at least 60 μg protein each, that were labelled with a 10-plex TMT kit (Supplementary Fig. 4, left). LC-SPS-MS[3] analysis of the U-2 OS LOPIT-DC fractions resulted in identification of 9386 protein groups after replicate merging and, following initial processing and missing value removal, 6837 protein groups with a full reporter ion series across all fractions and replicates remained (Supplementary Table 3)[50,66].

Principal component analysis (PCA) was used to visualise the protein profiles across the main sources of variance in our LOPIT-DC data. PCA is a dimensionality reduction method that transforms the original continuous multi-dimensional data into a set of uncorrelated variables (principal components), such that the first principal component accounts for as much variability in the data as possible and each succeeding component explains the greatest variance possible under the constraint that it be orthogonal to the preceding components. PCA is extremely useful for the visualisation of quantitative proteomics data to check if there is any underlying structure (i.e. organelle separation) and identify any hidden patterns in the data (that may represent subcellular niches). Using LOPIT-DC we are able to distinguish 10 subcellular compartments (Fig. 2a); the mitochondrion, nucleus/chromatin, endoplasmic reticulum (ER), Golgi, proteasome, peroxisome, cytosol, plasma membrane (PM), lysosome and ribosome. We observe that principal components (PCs) 1 and 2 broadly separate the organelle marker proteins into three groups: (1) membranous organelles excluding

the nucleus, (2) nucleus/chromatin, ribosome and proteasome and (3) cytosol. Importantly, subcellular niches that seem to overlap in PCs 1 and 2 are separated in other dimensions. For example, the Golgi apparatus and PM exhibit overlapping distributions in PCs 1 and 2 but are cleanly separated along PC4 (Fig. 2a, right). Similarly, the nucleus/chromatin and proteasome clusters overlap in PCs 1 and 2 but are separated along PC3. LOPIT-DC offers good reproducibility between replicates with respect to protein yield per fraction (Supplementary Fig. 4, left) and subcellular resolution (Supplementary Figs. 1a, 2a, 3).

**Comparison with hyperLOPIT**. We previously presented a hyperLOPIT map of U-2 OS cells based on two replicates[38]. Here, we extend this dataset with a third replicate for a thorough comparison with LOPIT-DC. The U-2 OS hyperLOPIT experiments required on average $2.8 \times 10^8$ cells each to obtain at least 70 μg protein in each fraction with the exception of the first 5–7 fractions which were pooled for further analysis (Supplementary Fig. 4, right). As hyperLOPIT aims to achieve maximum overall resolution a more involved TMT labelling strategy is required, such that two TMT 10-plexes are used for each replicate to label all density gradient fractions plus cytosol- and chromatin-enriched samples[50] (Fig. 1, Supplementary Data 1, 2). Due to the large number of samples analysed during our hyperLOPIT experiments, the amount of missing values which arose throughout the analysis was higher compared to the LOPIT-DC data and so the final combined hyperLOPIT dataset contains fewer proteins; following quantitative LC-SPS-MS[3] analysis of all three hyperLOPIT replicates we identified 9558 protein groups which were reduced to 4883 after filtering and concatenating replicates (Supplementary Table 3).

PCA of the hyperLOPIT dataset (Fig. 2b) reveals separation of 12 subcellular niches, as this method is able to resolve the individual ribosome subunits as well as the chromatin and nucleus clusters. We can generally class these subcellular niches into four groups in the first two components: (1) cytosol and proteasome, (2) nucleus, chromatin and ribosomes, (3) mitochondrion and peroxisome and (4) secretory pathway organelles (lysosome, PM, ER and Golgi; Fig. 2b, Supplementary Fig. 2b). The positions of these subcellular niches relative to one another in the hyperLOPIT PCA plots are broadly similar to those of the groups present in the LOPIT-DC data with two major exceptions. Firstly, in the hyperLOPIT data, the peroxisome and mitochondrion are in close proximity in the first 5 PCs and do not separate well until PC6 and PC8 (Fig. 2b, right) whereas these organelles sit at opposite ends of PC2 in the LOPIT-DC dataset. This can be explained by the fact that the peroxisome and mitochondrion have similar densities but different size ranges (0.4–0.8 and 0.4–2.5 μM, respectively)[67] and therefore are more easily separated from each other by differential rather than density gradient-based ultracentrifugation. Secondly, the proteasome and cytosol are close together in the first 4 PCs in our hyperLOPIT data whereas they are separated along PC1 in the LOPIT-DC dataset. This is likely explained by subtle differences in the way the cytosolic fraction is obtained in LOPIT-DC and hyperLOPIT. In hyperLOPIT, soluble complexes including the proteasome remain with the cytosolic proteins in the supernatant during the crude membrane-cushioning step, whereas in LOPIT-DC the proteasome and other macromolecular complexes are pelleted during the final spin ($120,000 \times g$) while the cytosolic proteins are extracted from the final supernatant. Additionally, while PC2 separates the secretory pathway organelles, mitochondrion and peroxisome in both datasets, the cluster order is different. In summary, both methods efficiently resolve all major subcellular

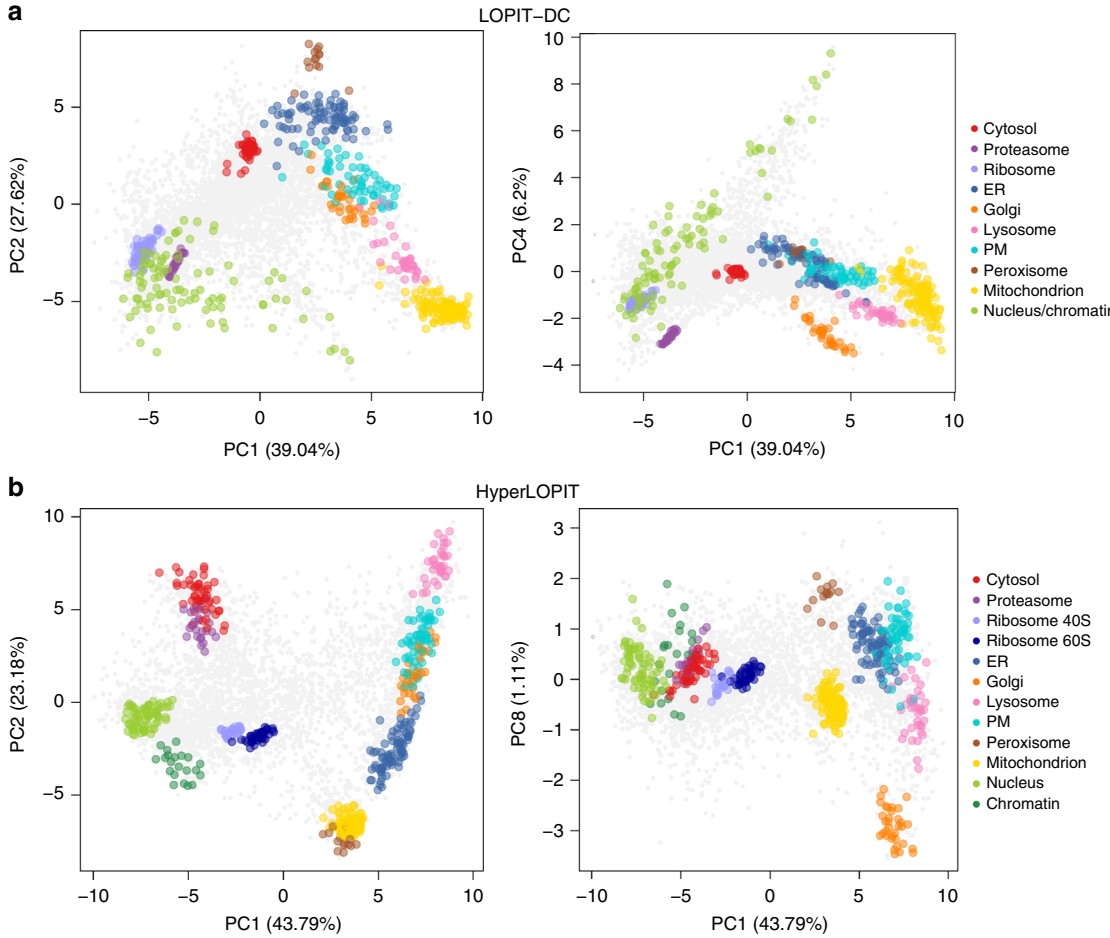

**Fig. 2** LOPIT-DC resolves all major subcellular niches. **a**, **b** Principal component analysis (PCA) projections for the LOPIT-DC (**a**) and hyperLOPIT (**b**) datasets. Subcellular marker proteins are highlighted as indicated and multiple principal components are presented to display resolution of all compartment clusters. Percentage variance explained by component is shown in parentheses

niches but the protein profile relationships in each compartment across the fractions are different due to the distinct fractionation techniques employed in each workflow.

**LOPIT-DC achieves high subcellular resolution**. Having established that LOPIT-DC is able to separate most major organelles, we proceeded to a quantitative comparison with hyperLOPIT. Firstly, we utilised QSep, a tool which quantifies the separation between a pair of organelle marker sets by comparing the intra-group and inter-group distances (see Methods and ref. [62]), to examine the resolution achieved by LOPIT-DC and hyperLOPIT (Fig. 3). For all comparisons presented in this manuscript we used the exact same subcellular marker list to analyse our two datasets. These markers were grouped into 10 or 12 subcellular classes to define the LOPIT-DC or hyperLOPIT data, respectively, with the ribosome 40S and ribosome 60S hyperLOPIT groups merged into one ribosome class for the LOPIT-DC dataset and, similarly, the nucleus and chromatin classes merged into a single nucleus/chromatin group in the LOPIT-DC data. This is because, as seen in Supplementary Fig. 5, the two ribosomal subunits are not well-separated from each other by LOPIT-DC. The same is true regarding the nucleus and chromatin clusters, with the latter being justified by the fact that, in contrast to our hyperLOPIT experiments, no separate chromatin-enriched fraction was added to our LOPIT-DC analysis.

Within the LOPIT-DC dataset, the smallest QSep distances are observed between the peroxisome/ER, ER/PM and lysosome/

mitochondrion pairs (Fig. 3a, left) in concordance with the LOPIT-DC PCA plot (Fig. 2a, left) where these organelles are positioned close together in PCA space, forming a continuum of clusters. In turn, the largest normalised distances are those between various organelles and the proteasome or ribosome, as also reflected by the LOPIT-DC PCA plot where the ribosome and proteasome are well-separated from the secretory pathway organelles and mitochondrion along PCs 1 and 2 (Fig. 2a). HyperLOPIT provides higher overall resolution compared to LOPIT-DC (Fig. 3a, right, 3c) but, importantly, some organelle pairs exhibit a higher QSep distance and the proteasome and cytosol are notably more separated from the other subcellular niches in the LOPIT-DC data (Fig. 3b). An additional comparison between the LOPIT-DC and hyperLOPIT data based on QSep distances when the LOPIT-DC dataset is annotated with 12 organelle classes is presented in Supplementary Fig. 6.

We next employed QSep to assess the overall subcellular resolution of our LOPIT-DC and hyperLOPIT data compared to publicly available spatial proteomics datasets. Having previously demonstrated that hyperLOPIT provides excellent resolution[62], we extended this comparison to include LOPIT-DC and observed that the U-2 OS LOPIT-DC dataset exhibits better subcellular separation relative to all other datasets with the exception of our U-2 OS hyperLOPIT data (Supplementary Fig. 7). This demonstrates the high quality of both of our datasets and the exceptionally high resolution achieved by LOPIT-DC.

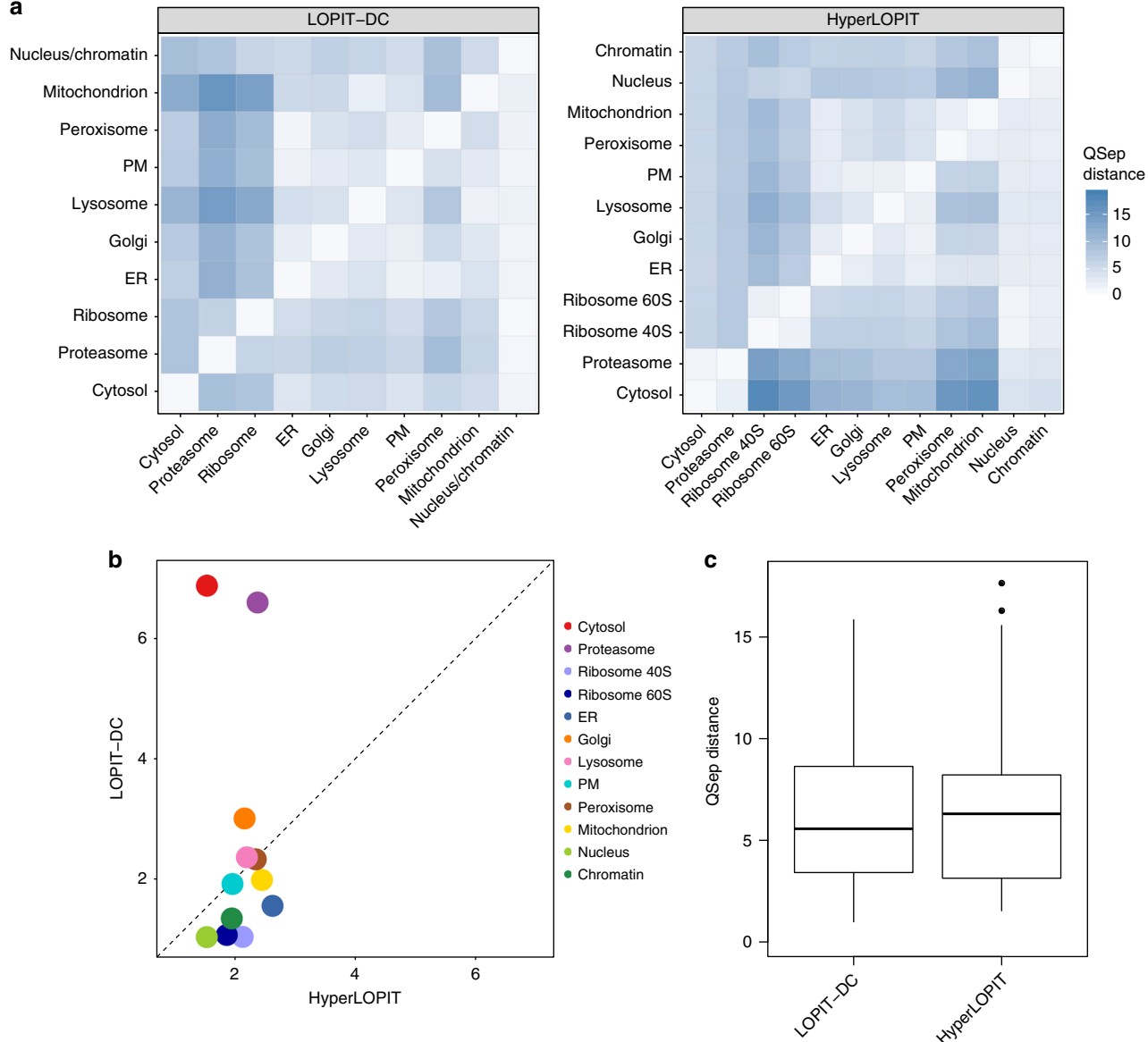

**Fig. 3** LOPIT-DC provides a high-resolution map of protein subcellular localisation. **a** QSep distances between all pairs of compartments in the LOPIT-DC (left) and hyperLOPIT (right) data. Ten organelle classes were used in the case of the LOPIT-DC data and 12 in the case of the hyperLOPIT dataset. **b** Comparison between LOPIT datasets of the minimum QSep distances for each of the 12 subcellular compartments. **c** Overall distribution of QSep distances for the LOPIT-DC and hyperLOPIT data. For each boxplot, the line in the middle of the box is the median value, the vertical size of the box represents the interquartile range (IQR) and the whiskers represent the extremes of the data (defined as those that do not exceed 1.5 × IQR from the middle of the data, and if no points exceed that distance, then the whiskers are simply the minimum and maximum values)

**LOPIT-DC and hyperLOPIT yield similar classifications**. After comparing our LOPIT-DC and hyperLOPIT datasets regarding subcellular resolution at the marker level, we expanded our characterisation to the level of protein localisation prediction. To classify each unlabelled protein to a unique subcellular location, we performed support vector machine (SVM)-based supervised machine learning using 10 organelle classes for the LOPIT-DC dataset and 12 for the hyperLOPIT data. As a first step in assessing classifier performance we examined the macro F1 scores (harmonic mean of precision and recall)[64] obtained after SVM parameter optimisation for each dataset, whereby a score of 1 indicates that the marker proteins are consistently assigned to the correct subcellular location by the algorithm[64]. The median macro F1 scores (acquired during SVM parameter optimisation iterations) were close to 1 for both datasets,

indicating that both LOPIT methods were able to correctly classify the subcellular markers with high generalisation accuracy (Fig. 4c).

Figure 4a, b show the U-2 OS LOPIT-DC and hyperLOPIT datasets, respectively, after SVM-based protein subcellular location classification followed by 5% FDR filtering. More proteins were assigned to a unique location in the LOPIT-DC data compared to the hyperLOPIT dataset but the proportion of classified proteins was slightly higher in the hyperLOPIT data (42%) as opposed to the LOPIT-DC dataset (35%) (Supplementary Table 4). We proceeded to a comparison between the SVM classifications obtained for the two datasets to explore the level of agreement achieved by our two distinct workflows and potential method-specific biases towards particular subcellular niches.

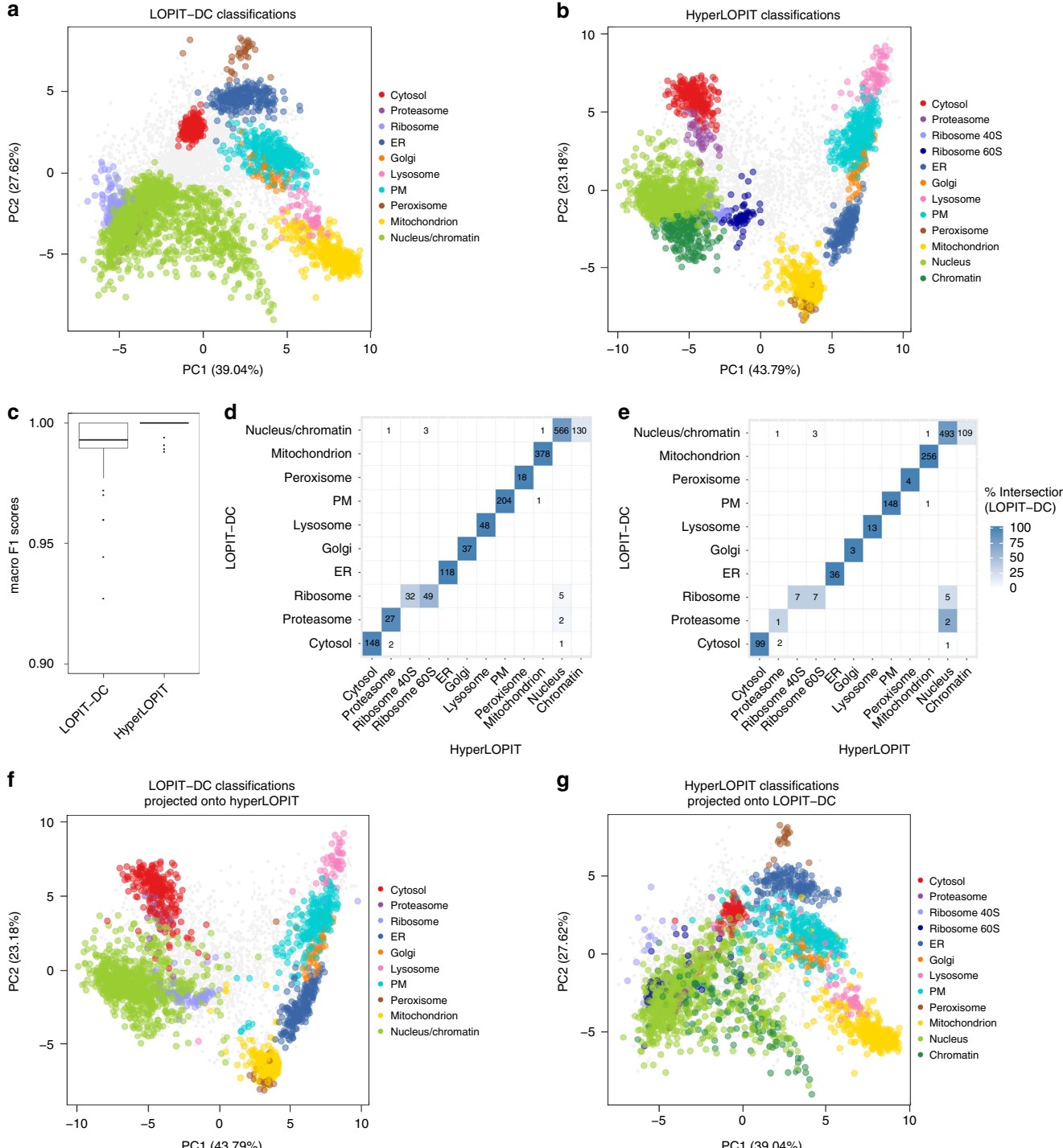

**Fig. 4** LOPIT-DC protein localisation classifications agree with hyperLOPIT. **a**, **b** PCA plots displaying SVM classification results after applying a 5% FDR cutoff for the LOPIT-DC (**a**) and hyperLOPIT (**b**) datasets. **c** Optimisation results (macro F1 scores) from SVM classification. For each boxplot, the line in the middle of the box is the median value, the vertical size of the box represents the interquartile range (IQR) and the whiskers represent the extremes of the data (defined as those that do not exceed 1.5 × IQR from the middle of the data, and if no points exceed that distance, then the whiskers are simply the minimum and maximum values). **d** Overlap between the LOPIT-DC and hyperLOPIT protein subcellular localisation assignments (including markers), where the LOPIT-DC dataset is classified using 10 marker classes and the percentage intersection is calculated relative to the number of proteins classified by compartment in the LOPIT-DC data. **e** As per **d** but excluding markers. **f**, **g** LOPIT-DC classifications projected onto the hyperLOPIT PCA (**f**) and vice versa (**g**). Proteins which were identified by both LOPIT-DC and hyperLOPIT are shown

We find 4162 proteins in common between the U-2 OS LOPIT-DC and hyperLOPIT datasets (including the 579 subcellular markers). Figure 4d provides an overview of the classification of these proteins to the classes used for SVM-

based protein localisation assignment including organelle markers (1771 proteins) and Fig. 4e depicts an overview of the classification excluding markers (1192 proteins). The majority of the chromatin and nucleus as well as the ribosome 40S and

ribosome 60S hyperLOPIT classifications were assigned to the nucleus/chromatin and ribosome niches, respectively, by LOPIT-DC. Importantly, only 16/1192 (1.3%) proteins classified in both datasets were assigned to different locations by LOPIT-DC and hyperLOPIT (Fig. 4d). Instead, the majority of classification disparities between our LOPIT-DC and hyperLOPIT data stem from cases where a protein was assigned to a unique subcellular niche in one dataset but remained unlabelled in the other; we provide a detailed analysis of the LOPIT-DC and hyperLOPIT unclassified proteomes in a later section. Finally, an additional assignment overlap overview for our two datasets including proteins which remained unlabelled and those that were present in one dataset but not the other is presented in Supplementary Fig. 8.

For most subcellular niches, the proteins classified by LOPIT-DC were also assigned to the same compartment by hyperLOPIT: ER, Golgi, lysosome, mitochondrion and peroxisome (100% agreement), nucleus/chromatin and PM (99%), cytosol (97%). The agreement was slightly lower for the ribosome, in which case 14/19 proteins were classified to either the 40S or 60S ribosome subunit by hyperLOPIT with the remaining proteins assigned to the nucleus (Fig. 4e). Finally, the overlap was lower for the proteasome, in which case only 1/3 proteins were also classified to the same subcellular niche by hyperLOPIT. Again, the remaining proteins were assigned to the nucleus in the hyperLOPIT dataset. This discrepancy suggests that, in rare instances, hyperLOPIT classifies proteins to the nucleus when they may belong to cytoplasmic complexes.

To further explore any potential biases in the protein localisation classifications, we projected the assignments of one LOPIT dataset onto the PCA of the other and vice versa. Plotting the LOPIT-DC classifications upon the hyperLOPIT PCA projections we see that the former form very similar clusters to those observed using the original hyperLOPIT assignments (Fig. 4f). However, the cytosol classifications spread slightly over the proteasome and nucleus clusters, matching our observation that 3% of the LOPIT-DC cytosol assignments are classified to the proteasome or nucleus by hyperLOPIT. Plotting the hyperLOPIT assignments upon the LOPIT-DC PCA projections we again observe similar clusters to the original, with the exception of the nucleus classifications which in this case seem more dispersed (Fig. 4g). Overall, the predictions obtained using LOPIT-DC are very similar to those acquired using hyperLOPIT (98.7% identical), with minor disagreements mainly involving proteins classified to the nucleus by hyperLOPIT.

**Transfer learning showcases the merit of method integration.** Transfer learning can be used for the meaningful integration of heterogeneous data sources in order to improve overall protein subcellular location classification[68]. The transfer learning approach is based on the integration of a primary experimental spatial proteomics dataset and an auxiliary dataset and results in higher generalisation accuracy than standard supervised machine learning workflows using a single information source[68]. We have previously shown that transfer learning is particularly useful for organelle classes not optimally resolved in the primary experimental data. We reasoned that the different biochemical fractionation approaches employed by LOPIT-DC and hyperLOPIT would convey individual strengths to each method such that combining the two approaches will yield the most accurate classification. Since the above analyses indicated that hyperLOPIT achieved higher overall resolution than LOPIT-DC, we used the hyperLOPIT data as the primary information source and the LOPIT-DC dataset as the auxiliary data. Fig. 5 (right) shows the distribution of the class-specific weights selected over 100 test

partitions of the transfer learning algorithm applied to the two datasets. These weights, one per organelle class, determine the proportion of primary and secondary data to be used for learning and range between 0 and 1[68]. A weight of 1 implies that all weight is given to the primary data source, meaning that the final result relies exclusively on the primary experimental dataset and ignores the auxiliary data source provided and vice versa for a weight of 0. A weight of 0.5 implies that both data sources are equally used during learning and so contribute equally to the final result. It is apparent that the weight distributions obtained over the 100 iterations performed on our two datasets reflect the resolution achieved by hyperLOPIT and LOPIT-DC (Fig. 5, left), with the distribution weights skewed towards 1 for 10/12 subcellular compartments suggesting that these organelles should be predominantly classified using the hyperLOPIT data. The exceptions to this are the cytosol, which was assigned a best weight of 0 in 78% of iterations, and the proteasome, which was assigned a best weight of 0 or 0.5 in 85% of iterations. This indicates that the LOPIT-DC data should be predominantly used to classify proteins to these compartments, reflecting the overlapping distributions exhibited by the cytosol and proteasome in the hyperLOPIT dataset and, in turn, their excellent separation from each other and all other organelles in the LOPIT-DC data. Finally, the macro F1 scores obtained after weight optimisation demonstrate that including the LOPIT-DC data in the classification leads to an increase in classifier prediction accuracy relative to the case of using the hyperLOPIT dataset alone (Fig. 5, right). In conclusion, these findings highlight the merit of integrating our two spatial proteomics methods to achieve optimal classification of proteins to organelles.

**More than half of the proteome is in multiple locations.** Sixty-five percent of proteins remained unclassified in the LOPIT-DC data after SVM-based subcellular location prediction and subsequent 5% FDR filtering. This compares to 58% for the hyperLOPIT dataset. There are multiple explanations for proteins remaining unclassified, including that they: (1) reside in more than one subcellular compartment, (2) associate with dynamic components, (3) actively traffic between different subcellular locations, (4) belong to subcellular structures for which no known markers were included in the analysis and/or (5) were inaccurately quantified across the fractions leading to erroneous classification. To explore the above possibilities, we performed further exploratory analyses of the LOPIT-DC and hyperLOPIT unclassified proteomes.

As an initial assessment of the subcellular distribution of our unclassified proteins we utilised the immunofluorescence-based, U-2 OS cell-specific protein subcellular localisation information available as part of the Cell Atlas database[38]. The clearest over-represented Cell Atlas localisation for both the LOPIT-DC and hyperLOPIT unassigned proteins is the cytosol (Fig. 6a). This is expected, as many known translocators are soluble cytosolic proteins capable of trafficking to different subcellular compartments to exert their function(s)[69].

Notably, the unassigned proteins in both datasets show a clear under-representation for proteins localised to the mitochondria, indicating that mitochondrial proteins are more likely to be classified. This may be because mitochondrial proteins, having a specific import signal sequence, are more definitively localised to the mitochondria and therefore less likely to migrate to other subcellular compartments.

To gain additional insights into the multilocalising proteome captured by our two distinct workflows, we next performed functional Gene Ontology (GO)[70] term enrichment analysis on the proteins which remained unassigned in the LOPIT-DC and

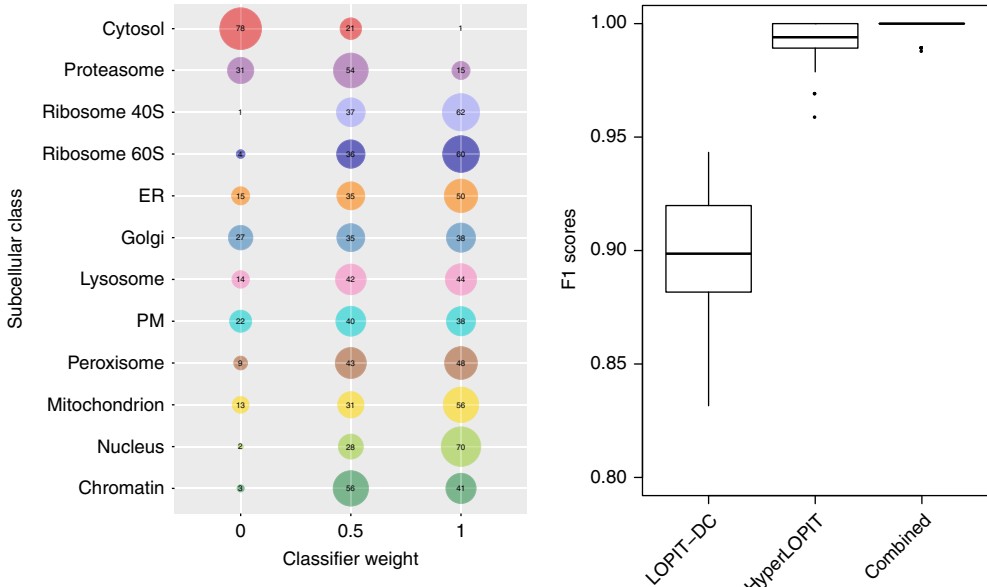

**Fig. 5** Transfer learning using the hyperLOPIT and LOPIT-DC data as main and auxiliary sources. (left) Optimisation of transfer learning weighting. Numbers in bubbles represent frequencies obtained for each weight over 100 iterations; (right) F1 scores for the LOPIT-DC dataset, hyperLOPIT dataset and the combination of the two. For each boxplot, the line in the middle of the box is the median value, the vertical size of the box represents the interquartile range (IQR) and the whiskers represent the extremes of the data (defined as those that do not exceed 1.5 × IQR from the middle of the data, and if no points exceed that distance, then the whiskers are simply the minimum and maximum values)

hyperLOPIT data. These proteins are enriched with GO terms related to the cytosol/cytoplasm, endomembrane system and vesicle-mediated transport (Fig. 6b, c), therefore supporting our Cell Atlas-related observations according to which the LOPIT-DC and hyperLOPIT unclassified proteins are enriched in translocators.

**LOPIT-DC achieves suborganellar and isoform resolution**. Once we confirmed that LOPIT-DC achieves high resolution at the level of whole organelles and explored the multilocalising proteome captured by this method, we proceeded to examine whether this technique can additionally resolve suborganellar structures. As demonstrated in Fig. 7a, LOPIT-DC resolves sub-organellar compartments with superb agreement with hyperLO-PIT. For example, proteins that belong to the ER lumen, ER membrane and cis-Golgi display distinct profiles and ERGIC-cis Golgi markers are situated between the ER and Golgi clusters in both the LOPIT-DC and hyperLOPIT PCA plots (Fig. 7a). We also looked at structures that are difficult to resolve in bio-chemical fractionation-based spatial proteomics experiments and which were not part of our classification marker set, such as the actin cytoskeleton. As expected due to the broad connectivity of cytoskeletal components with most subcellular structures, actin-binding proteins are mainly distributed to the unassigned area of both LOPIT PCA plots (Supplementary Fig. 9).

We next sought to investigate the location of large protein complexes. The majority of complexes we examined are localised to discrete positions in an organelle and exhibit identical distributions in the two LOPIT datasets (Fig. 7b, Supplementary Fig. 10). For example, in both the LOPIT-DC and hyperLOPIT datasets, the SUMO-activating enzyme and COP9 signalosome complexes are co-localised within the cytosolic cluster, the signal peptidase and OST (N-oligosaccharyl transferase) complexes are both situated in the ER cluster, the snRNPs and Origin Recognition Complex (ORC) are both positioned within the nuclear cluster and the ATP synthase complex is located within the mitochondrial cluster, indicating that LOPIT-DC is able to

resolve organellar substructures including macromolecular complexes. Additionally, we present U-2 OS cell-specific subcellular localisation information, taken from the Cell Atlas database, corresponding to individual members of four of the protein complexes (ATP synthase, OST, ORC and snRNPs) illustrated in Fig. 7b (Supplementary Fig. 11). Notably, the vast majority (15/17 proteins examined) of these annotations agree with the localisa-tion of these proteins in our LOPIT data. Interestingly, in the rare cases where there is disagreement between the LOPIT and Cell Atlas localisations (in the cases of UniProt accessions P46977 and P62316), we find that these specific annotations are classed as uncertain in the Cell Atlas database. This is in support of the observation by Thul et al.[38] that the agreement between hyperLOPIT- and Cell Atlas-based subcellular localisation assignments varies for the different reliability tiers of the Human Protein Atlas project.

We also observe that the majority of the individual signalling pathway constituents we examined are found in the same subcellular niche in the two LOPIT datasets. Since our experiments were performed using an osteosarcoma cell line, we explored pathways which play critical roles in cancer (Supplementary Figs. 12–18). For example, we identified 10 components of the p53 signalling pathway which controls DNA replication and cell division and has been implicated in many cancers[71]. As exhibited in Fig. 7c, 3/10 p53 signalling proteins were classified to the same organelle in both LOPIT datasets (PM, mitochondrion or nucleus), three were only classified in one dataset but are positioned close to the same organelle in the PCA plot of the other dataset and four remained unclassified by both LOPIT workflows but are situated close to the same subcellular niche in the PCA plots of both the LOPIT-DC and hyperLOPIT data.

Finally, we also explored the distribution of individual protein isoforms in our two LOPIT datasets. As seen in Supplementary Fig. 19, we could identify six examples where two different isoforms of a protein were detected in both the LOPIT-DC and hyperLOPIT data. In every case, the isoforms are mapped to the same subcellular niche in both datasets. Intriguingly, the

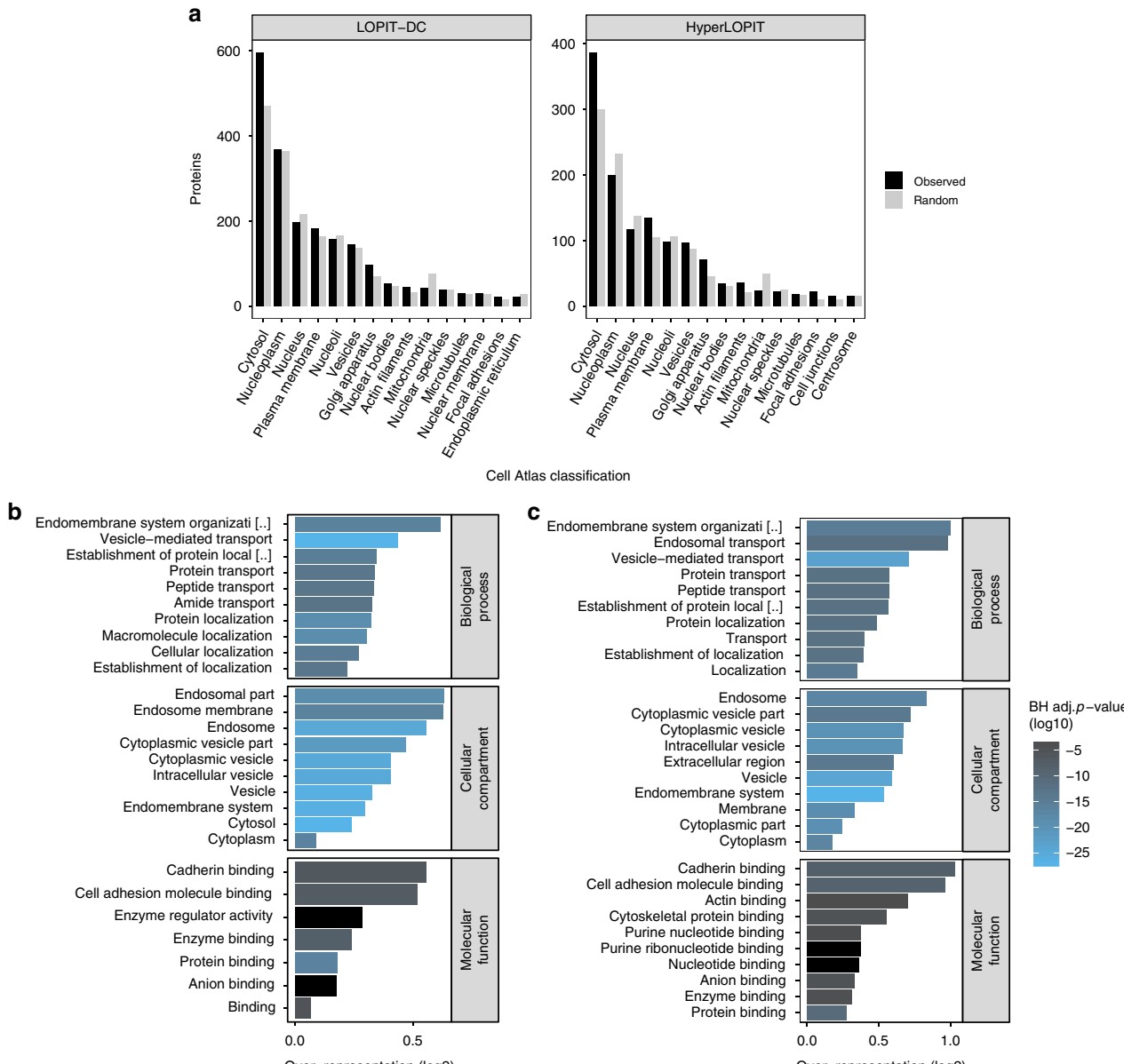

**Fig. 6** The LOPIT unclassified proteome reveals proteins with multiple locations. **a** Cell Atlas database-derived subcellular location for proteins that remained unclassified in the LOPIT-DC and hyperLOPIT data. Grey bars represent the expectation from a random selection of the same number of proteins. Only the top 15 most frequent localisations are shown. **b**, **c** GO terms over-represented in the unclassified proteins for LOPIT-DC (**b**) and hyperLOPIT (**c**). Only the top 10 most over-represented terms per category are shown. BH adj. *p*-value = Benjamini-Hochberg[78] adjusted *p*-value for over-representation

canonical isoform of MAP4 (Microtubule-Associated Protein 4; P27816–1) was classified to the nucleus by both LOPIT approaches, in agreement with its published role in the coordination of spindle orientation[72]. In contrast, MAP4 isoform 4 (P27816–4) remained unclassified but is situated at the border between the nuclear and ribosomal clusters in the PCA plots of both datasets. UniProt further identifies this protein as part of the cytoskeleton, cytosol, PM and extracellular or secretory vesicles. Our data indicate that MAP4 localisation is likely isoform-specific and thus these isoforms might perform different microtubule-related functions.

## Discussion

The hyperLOPIT workflow is a well-established method to generate proteome-wide, high-resolution maps of protein subcellular localisation[50] and has been applied to the study of several different biological systems[41–46,48–50]. The U-2 OS hyperLOPIT dataset presented here is the largest human (hyper)LOPIT dataset reported thus far and the most highly resolved MS-based human subcellular map published to date (Supplementary Fig. 7). However, hyperLOPIT can be time-consuming and labour- and resource-intensive. In this manuscript, we have demonstrated that retaining most elements of the hyperLOPIT protocol but replacing the density gradient with differential centrifugation addresses these issues, achieving a near-hyperLOPIT level of resolution and protein subcellular localisation classification results which strongly agree with those provided by hyperLOPIT.

We have provided a quantitative comparison between the spatial resolution and protein subcellular localisation assignment output achievable applying density gradient- or differential centrifugation-based spatial proteomics methods to the same cell

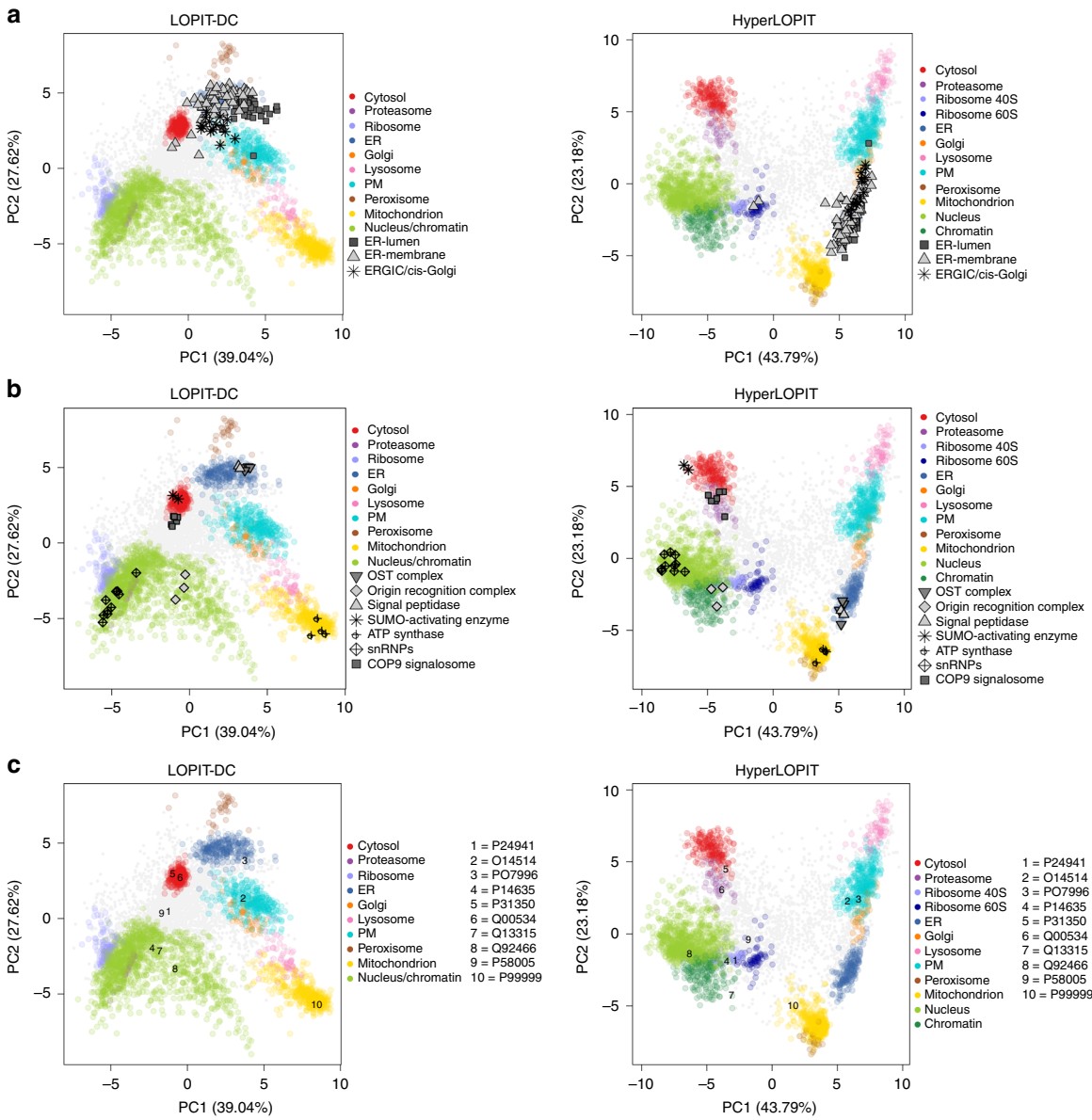

**Fig. 7** Suborganellar structures, complexes and pathways in the LOPIT-DC and hyperLOPIT data. **a** ER-lumen, ER-membrane and ERGIC/cis-Golgi markers plotted upon the LOPIT-DC and hyperLOPIT datasets with assigned proteins. **b** COP9 signalosome, snRNPs, ATP synthase, signal peptidase, SUMO-activating enzyme, OST complex and Origin Recognition Complex plotted upon the LOPIT-DC and hyperLOPIT datasets with assigned proteins. **c** Ten proteins involved in p53 signalling plotted upon the LOPIT-DC and hyperLOPIT datasets with assigned proteins

line and by the same laboratory. While density gradient-based cell fractionation in hyperLOPIT achieves higher overall resolution, differential centrifugation-based fractionation in LOPIT-DC appears to resolve the cytosolic and proteasomal compartments more effectively. In addition, LOPIT-DC provides much greater separation between the mitochondrion and peroxisome compared to hyperLOPIT. Notably, one of the major advantages of the hyperLOPIT workflow is the extra layer of resolution provided by an additional chromatin preparation which enables separation between chromatin-associated and other nucleoplasm proteins. Whilst our version of LOPIT-DC does not include this preparation and is therefore not expected to separate these proteins, we nonetheless do observe a level of separation to a lesser degree (Supplementary Fig. 5). Where further resolution of chromatin-associated proteins or of any other additional subcellular compartment is desirable, separate preparations can be

added to the LOPIT-DC protocol via the inclusion of an 11th TMT label in the analysis.

In conclusion, both of our LOPIT workflows accomplish high overall resolution. The choice regarding which one to use depends on the biological question in mind as well as the amount of starting material, time and resources available. Where resources are not limited, hyperLOPIT provides the maximum possible overall subcellular resolution but in cases of starting material, time or financial constraints the simpler and quicker LOPIT-DC protocol can offer a great all-in-one alternative. This workflow may be more suitable for time-course experiments or cases where sample processing speed is important to retain biological information. In addition, in situations where the focus of the study is the mitochondrion, peroxisome, cytosol or proteasome there might be an argument in favour of using LOPIT-DC instead of (or in combination with) hyperLOPIT, due to the

efficient and complementary separation of these compartments achieved by this workflow. As our findings demonstrate, both methods yield reliable, comparable results and can be utilised in the context of dynamic studies or for the mapping of features such as post-translational modifications, protein interactions or isoform-specific functions. Importantly, our U-2 OS LOPIT-DC and hyperLOPIT data are the highest-resolution MS-based spatial proteomics maps created using human cells to date; these datasets provide a snapshot of the structural organisation of U-2 OS cells and can serve as a reference for future studies on human protein subcellular localisation and its relationship to protein function.

## Methods

**Cell culture.** The U-2 OS human osteosarcoma cell line (RRID: CVCL_0042) was a generous gift from Emma Lundberg (SciLifeLab Stockholm and School of Biotechnology, KTH). Cells were grown at 37 °C and 5% $CO_2$ in McCoy's 5A medium (Sigma) supplemented with sodium bicarbonate, 10% foetal bovine serum (Biosera) and 1% GlutaMax$^{TM}$ (Life Technologies), without antibiotics.

**Sample preparation.** LOPIT-DC utilises the same cell lysis approach as hyperLOPIT which involves a gentle isotonic lysis buffer that keeps the organelles as intact as possible while the cells are lysed in a ball-bearing cell homogeniser. This step shows excellent reproducibility and can be optimised for a variety of cell or tissue types (http://www.isobiotec.com/cell-type.html). The cell lysis stage is important in both LOPIT-DC and hyperLOPIT as inefficient lysis results in sub-optimal organelle recovery which leads to low protein yields during later steps, reducing overall efficiency. For LOPIT-DC, inefficient lysis would also mean generation of large microsomal particles which would sediment during the initial centrifugation steps.

U-2 OS cells were harvested by trypsinisation, washed with PBS (pH 7.4) three times and resuspended in a gentle lysis buffer (0.25 M sucrose, 10 mM HEPES pH 7.4, 2 mM EDTA, 2 mM magnesium acetate, protease inhibitors). They were then lysed in a ball-bearing homogeniser (isobiotec) using a pair of 1 mL syringes. Each mL of the cell suspension was passed through the homogeniser chamber 15 times and a 12 µm ball-bearing clearance size was used. In the case of hyperLOPIT, the combined cell lysate was then treated with 500 U of the nuclease Benzonase (Sigma-Aldrich) for 20 min at room temperature and for a further 10 min at 4 °C in order for its viscosity to be reduced. All lysates were spun three times at 200 × g, 5 min, 4 °C to remove any unlysed cells.

For hyperLOPIT, a chromatin extraction step was performed in parallel using approximately 5–15 million cells (10–20% of the total number of cells used per experiment). In detail, the cells were pelleted at 200 × g, 5 min, 4 °C and resuspended in 5 mL of chromatin buffer A (10 mM HEPES pH 7.9, 10 mM KCl, 1.5 mM MgCl₂, 340 mM sucrose, 10% (v/v) glycerol, 1 mM DTT, protease inhibitors). Triton X-100 was added to the cell suspension to a total concentration of 0.1% (v/v). The cell suspension was then gently mixed by inversion and left on ice for 8 min. Next, the sample was centrifuged at 1300 × g, 5 min, 4 °C. The supernatant was discarded and the crude nuclear pellet resuspended in another 5 mL of chromatin buffer A and centrifuged again at 1300 × g, 5 min, 4 °C. This crude nuclear fraction was lysed by adding 5 mL of chromatin buffer B (3 mM EDTA, 0.2 mM EGTA, 1 mM DTT, protease inhibitors), gently mixing by inversion and incubating the sample on ice for 30 min. Next, the sample was centrifuged at 1700 × g, 5 min, 4 °C, the supernatant discarded and the pellet resuspended in a further 5 mL of chromatin buffer B. The washed pellet was centrifuged again at 1700 × g, 5 min, 4 °C, the supernatant discarded and the pellet stored at −80 °C.

**HyperLOPIT subcellular fractionation.** Samples for hyperLOPIT were fractionated using an iodixanol density gradient[38,63]. In detail, a cell lysate from approximately 280 million cells per average experiment was first separated into a cytosol-enriched and a crude membrane fraction using 6 and 25% (w/v) iodixanol-containing solutions and centrifugation at 100,000 × g, 90 min, 4 °C in an Optima L-80 XP Beckman ultracentrifuge and a SW55Ti rotor (maximum acceleration, minimum deceleration). The supernatant was stored and the membrane fractions situated at the interface of the iodixanol layers collected, diluted 5-fold with lysis buffer and centrifuged at 200,000 × g, 60 min, 4 °C using the same ultracentrifuge and rotor in order to remove any residual cytosolic contamination. The membranes were then resuspended in 25% (w/v) iodixanol, gently homogenised using a 1-mL Dounce homogeniser (Wheaton), underlaid beneath a linear gradient of 8%, 12%, 16 and 18% (w/v) iodixanol solutions and fractionated by centrifugation at 100,000 × g, 8 h, 4 °C using a VTi65.1 fixed-angle vertical rotor (replicate 1) or an NVT65 fixed-angle near-vertical rotor (replicates 2 and 3) with maximum acceleration and minimum deceleration. After ultracentrifugation, approximately 22 0.5-mL fractions were collected using an Auto Densi-Flow peristaltic pump fraction collector with a meniscus tracking probe (Labconco). These subcellular fractions were diluted with lysis buffer, centrifuged four times at 100,000 × g, 1 h, 4 °C using an Optima MAX-XP Beckman benchtop ultracentrifuge and a TLA-55

rotor in order to allow the membranes to pellet out of the iodixanol and stored at −80 °C. During all the above steps, the samples were kept on ice at all times in order to avoid membrane degradation.

The cytosol-enriched supernatant was precipitated with five volumes of cold acetone overnight at −20 °C. The obtained precipitated pellet and membrane pellets were resolubilised using 8 M urea, 0.2% SDS, 50 mM HEPES pH 8.5 and sonication. The fractions were centrifuged (16,000 × g, 10 min, 4 °C) to remove any insoluble material. Protein concentration was measured using the BCA protein assay kit (Thermo Fisher Scientific) according to the manufacturer's instructions.

Sixty to seventy micrograms of protein per fraction were reduced with a final concentration of 2.5 mM TCEP (Tris(2-carboxyethyl)phosphine) for 1 h at room temperature and alkylated with a final concentration of 5 mM MMTS (Methyl methanethiosulfonate) for 1 h at room temperature. The samples were then diluted 10-fold with 50 mM HEPES pH 8.5 (at this point it was ensured that the pH of each sample was >8) and digested with sequencing-grade trypsin (Promega) at an enzyme/protein ratio of 1/40 for 1 h at 37 °C. A second trypsinisation step was performed overnight (no longer than 16 h) at 37 °C and at the same enzyme/protein ratio. The next day, the trypsin digests were briefly centrifuged (16,000 × g, 10 min, 4 °C) to remove any insoluble material and subsequently reduced to dryness in a refrigerated vacuum centrifuge with a cold trap (Labconco). Each sample was then resuspended in 100 µL of 50 mM HEPES pH 8.5 (at this point it was ensured that the pH of each sample was >8), centrifuged at 16,000 × g, 10 min, 4 °C to remove any insoluble material and labelled with TMT isobaric tagging reagents (Thermo Fisher Scientific). More specifically, the TMT tags were equilibrated to room temperature and each resuspended in 85 µL of LC-MS-grade acetonitrile. For our hyperLOPIT samples, the tags from a TMT 10-plex kit were split in half (essentially rendering the labelling scheme a 20-plex experiment) and used to label all the membrane fractions (some were pooled to ensure adequate protein amounts) as well as the cytosol-enriched and chromatin-enriched samples for 2 h at room temperature, on a shaking platform. The labelling reaction was quenched by adding 8 µL of 5% (w/v) hydroxylamine (which was prepared in 100 mM HEPES pH 8.5) to every sample followed by a 30-min incubation at room temperature and on a shaker, and a further 100 µL of deionised water followed by a 1-h incubation at 4 °C. After labelling and quenching, peptides were pooled into 10-plexes and reduced to dryness by vacuum centrifugation. Three TMT 10-plex kits were used to label three biological replicates.

The combined, TMT-labelled samples were then cleaned using C18 SepPak cartridges (Waters) to remove salts and other substances which could interfere with downstream peptide processing. In detail, each sample was resuspended in approximately 5 mL of 0.1% (v/v) TFA (Trifluoroacetic acid; prepared in HPLC-grade water) and the pH of the solution adjusted to <3. The C18 SepPak cartridges were equilibrated using 2 mL of 100% (v/v) acetonitrile followed by 2 mL of 70% (v/v) acetonitrile and 0.05% (v/v) acetic acid (in HPLC-grade water), 2 mL of 0.05% (v/v) acetic acid (in HPLC-grade water) and 4 mL of 0.1% (v/v) TFA. The TMT-labelled peptides were slowly loaded onto the cartridges and the columns washed with 4 mL of 0.1% (v/v) TFA followed by 4 mL of 0.05% (v/v) acetic acid. The peptides were eluted from the cartridges using 2 mL of the 70% (v/v) acetonitrile + 0.05% (v/v) acetic acid solution and subsequently reduced to dryness by vacuum centrifugation.

The samples were then fractionated using high-pH reverse phase chromatography. In detail, the TMT-labelled peptide samples were resuspended in 100 µL of 20 mM ammonium formate pH 10 (Buffer A) and the pH of the solution was adjusted so that it be >9. The total volume of each sample was injected onto an Acquity UPLC BEH C18 column (2.1-mm i.d. × 150-mm; 1.7-µm particle size) on an Acquity UPLC System with a diode array detector (Waters) and the peptides were eluted from the column using a linear gradient of 4–60% (v/v) acetonitrile in 20 mM ammonium formate pH 10 over 50 min and at a 0.244 mL/min flow rate (with a total run time of 75 min). The gradient was set up as follows: 0 min–95% Buffer A–5% Buffer B (20 mM ammonium formate pH 10 + 80% (v/v) acetonitrile), 10 min–95% Buffer A–5% Buffer B, 60 min–25% Buffer A–75% Buffer B, 62 min–0% Buffer A–100% Buffer B, 67.5 min–0% Buffer A–100% Buffer B, 67.6 min–95% Buffer A–5% Buffer B. Approximately 40–50 1-min fractions, representing peak peptide elution, were collected from the moment that the peptides began to elute and were reduced to dryness by vacuum centrifugation shortly thereafter. For downstream MS analysis, the fractions corresponding to each TMT 10-plex set were orthogonally combined into 18–22 samples by combining pairs of fractions which eluted at different time points during the gradient.

**LOPIT-DC subcellular fractionation.** Samples for LOPIT-DC were fractionated using differential centrifugation (Supplementary Table 2). Importantly, among the novel aspects of the LOPIT-DC workflow is the extended centrifugation scheme aiming to increase the number of fractions across which to determine correlation profiles and hence improve resolution and the ability to capture all subcellular niches in a single experiment. A cell lysate from approximately 70 million cells per average experiment was separated into 10 fractions using the Eppendorf 5804 R for the first centrifugation step and the Optima$^{TM}$ MAX-XP Beckman benchtop ultracentrifuge with the TLA-55 rotor for the rest. All pellets and the last supernatant were stored at −80 °C.

The final supernatant was precipitated with five volumes of cold acetone overnight at −20 °C. The obtained precipitated pellet and membrane pellets were resolubilised in 8 M urea, 0.15% SDS and 50 mM HEPES pH 8.5. Protein concentration was measured using the BCA protein assay according to the manufacturer's instructions.

Fifty micrograms of protein per fraction were reduced, alkylated, digested and TMT-labelled in exactly the same manner as for the hyperLOPIT experiments described above. One TMT 10-plex kit was used to label all the membrane and cytosol-enriched fractions in three biological replicates.

After labelling, peptides were pooled into 10-plexes, cleaned with C18 SepPak cartridges and fractionated using high-pH reverse phase chromatography as described above for our hyperLOPIT experiments. The resulting fractions corresponding to each TMT 10-plex set were orthogonally combined into 18 samples for downstream MS analysis.

**SPS-MS³ on the Orbitrap Fusion Lumos.** All MS runs were performed on an Orbitrap Fusion™ Lumos™ Tribrid™ instrument coupled to a Dionex Ultimate™ 3000 RSLCnano system (Thermo Fisher Scientific).

Briefly, each of the fractionated samples was resuspended in 30 μL of 0.1% (v/v) formic acid. Approximately 1 μg of peptides was loaded per injection for LC-MS/MS analysis.

The nano-flow liquid chromatography method for LC-MS/MS was set as follows: Solvent A was 0.1% (v/v) formic acid. Solvent B was 80% (v/v) acetonitrile + 0.1% (v/v) formic acid. Loading solvent was 0.1% (v/v) formic acid. Peptides were loaded onto a micro precolumn (300-μm i.d. x 5 mm, particles were C18 PepMap 100, 5-μm particle size, 100-Å pore size, Thermo Fisher Scientific) using the loading pump for 3 min. After this, the valve was switched from load to inject. Peptides were separated on a Proxeon EASY-Spray column (PepMap RSLC C18, 50-cm x 75-μm i.d., 2-μm particle size, 100-Å pore size, Thermo Fisher Scientific) using a 2–40% (v/v) gradient of acetonitrile + 0.1% (v/v) formic acid at 300 nL/min over 93 min. A wash step (90% solvent B for 5 min) was included, followed by re-equilibration. The total run time was 120 min.

The MS workflow parameters were set as follows using the Method Editor in XCalibur v3.0.63 (Thermo Fisher Scientific) for the SPS-MS³ acquisition method:

Detector type: Orbitrap—Resolution: 120,000—Mass range: Normal—Use quadrupole isolation: Yes—Scan range: 380–1,500—RF lens: 30%—AGC target: 4e5—Max inject time: 50 ms—Microscans: 1—Data type: Profile—Polarity: Positive—Monoisotopic peak determination: Peptide—Relax restrictions when too few precursors are found: Yes—Include charge state(s): 2–7—Exclude after n times: 1—Exclusion duration (s): 70—Mass tolerance (p.p.m.): Low: 10; high: 10—Exclude isotopes: Yes—Perform dependent scan on single charge state per precursor only: Yes—Intensity threshold: 5.0e3—Data-dependent mode: Top speed —Number of scan event types: 1—Scan event type 1: No condition—MSn level: 2—Isolation mode: Quadrupole—Isolation window (*m/z*): 0.7—Activation type: CID —CID collision energy (%): 35—Activation Q: 0.25—Detector type: Ion trap—Scan range mode: Auto—*m/z*: Normal—Ion trap scan rate: Turbo—AGC target: 1.0e4—Max inject time (ms): 50—Microscans: 1—Data type: Centroid—Mass range: 400–1200—Exclusion mass width: *m/z*: Low: 18; high: 5—Reagent: TMT—Precursor priority: Most intense—Scan event type 1: No condition—Synchronous precursor selection: Yes—Number of precursors: 10—MS isolation window: 0.7—Activation type: HCD—HCD collision energy (%): 65—Detector type: Orbitrap—Scan range mode: Define *m/z* range—Orbitrap resolution: 60,000—Scan range (*m/z*): 100–500—AGC target: 1.0e5—Max inject time (ms): 120—Microscans: 1—Data type: Profile; AGC, automatic gain control; HCD, higher-energy collisional dissociation; CID, collision-induced dissociation.

An electrospray voltage of 2.1 kV was applied to the eluent via the electrode of the EASY-Spray column. The mass spectrometer was operated in positive ion data-dependent mode for SPS-MS³. The total run time was 120 min.

**Raw data processing and quantification.** Raw files from both the LOPIT-DC and hyperLOPIT experiments were processed the same way with Proteome Discoverer v1.4 (Thermo Fisher Scientific) using the Mascot server v2.3.02 (Matrix Science). The SwissProt sequence database for *Homo sapiens* (canonical and isoform, 42,118 sequences, downloaded on 04/11/2016) was used along with common contaminants from the common Repository of Adventitious Proteins (cRAP) v1.0 (48 sequences, adapted from the Global Proteome Machine repository, https://www.thegpm.org/crap/). Precursor and fragment mass tolerances were set to 10 ppm and 0.6 Da, respectively. Trypsin was set as the enzyme of choice and a maximum of 2 missed cleavages were allowed. Static modifications were: methylthio (C), TMT6plex (N-term) and TMT6plex (K). Dynamic modifications were: oxidation (M) and deamidated (NQ). Percolator was used to assess the false discovery rate (FDR) and only high-confidence peptides were retained. Additional data reduction filters were: peptide rank = 1 and ion score >20.

Quantification at the MS³ level was performed within the Proteome Discoverer workflow using the centroid sum method and an integration tolerance of 2 mmu. Isotope impurity correction factors were applied. Each raw peptide-spectrum match (PSM) reporter intensity was then divided by the sum of all intensities for that PSM (sum normalisation). Protein grouping was carried out according to the minimum parsimony principle and the median of all sum-normalised PSM ratios belonging to each protein group was calculated as the protein group quantitation

value. Only proteins with a full reporter ion series were retained. Additionally, three of the 60 TMT channels present in the final hyperLOPIT dataset possessed extremely low ion intensity profiles and were excluded from downstream data analysis to minimise background noise in the data. Finally, proteins identified as cRAP were also removed for downstream analysis.

**SVM-based prediction of protein localisation.** Data analysis was performed using the R[73] Bioconductor[74] packages MSnbase v2.6.1[75] and pRoloc v1.21.9[33] as described in[64]. Briefly, 579 manually curated marker proteins were used to define 12 subcellular locations: cytosol, proteasome, nucleus, chromatin, ribosome 40S, ribosome 60S, peroxisome, mitochondrion, lysosome, Golgi apparatus, plasma membrane (PM) and endoplasmic reticulum (ER) (Supplementary Data 1, 2). These markers were chosen based on information gleaned from UniProt and the literature. Care was taken not to choose markers based on previous hyperLOPIT studies on U-2 OS cells in case this gave rise to self-prophesying subcellular locations. These constitute our core organelle markers, proteins known to localise to one specific subcellular niche. Supervised machine learning using a support vector machine (SVM) classifier with a radial basis function kernel was employed in order to predict the localisation of unlabelled proteins. For the LOPIT-DC data, classification was performed using 10 marker classes, in which case the pairs nucleus/chromatin and ribosome 40S/ribosome 60S were merged to form single classes. Following the protocol in[63], one hundred rounds of fivefold cross-validation were employed (creating five stratified test/train partitions) to estimate algorithmic performance. This protocol features an additional round of cross-validation on each training partition to optimise the free parameters of the SVM, sigma and cost, via a grid search. Based on the best F1 score (the harmonic mean of precision and recall), the best sigma and cost for the hyperLOPIT dataset were 0.01 and 8, respectively. The best sigma and cost for the LOPIT-DC dataset were 0.01 and 16, respectively. All proteins assigned to a specific subcellular niche by SVM-based classification were ordered according to their SVM scores and a threshold was set to achieve a 5% FDR based on agreement with the UniProt and Gene Ontology databases as well as the literature.

**Data integration by transfer learning.** To show the complementary nature of the hyperLOPIT and LOPIT-DC methods at predicting subcellular location, we applied a transfer learning algorithm[68]. The transfer learning method allows one to integrate heterogeneous datasets (a primary and an auxiliary dataset) for optimal classification. Following the protocol described in ref. [68], the hyperLOPIT dataset was used as the primary source and the LOPIT-DC as the auxiliary source. Labelled marker proteins common in both datasets were extracted and the hyperLOPIT and LOPIT-DC quantitative protein profiles were used as input to the *k*-nearest neighbour transfer learning (knntl) algorithm. Three different experiments were conducted: (1) using the hyperLOPIT data only, (2) using the LOPIT-DC data only and finally (3) using both hyperLOPIT and LOPIT-DC data. As per the SVM classifier, one hundred rounds of fivefold cross-validation were used to estimate the optimal number of nearest neighbours for the *k*-nearest neighbour (*k*-NN) classifier. These were 5 and 5 for the hyperLOPIT and LOPIT-DC datasets, respectively. In the *k*-NN transfer learning framework we also need to estimate the parameter theta which is a vector of weights (one per organelle) used to control the amount of primary (hyperLOPIT) and auxiliary (LOPIT-DC) data to use in classification. We tested all weight combinations of 0, 0.5 and 1 for each organelle class and performed 100 iterations of cross-validation to determine the optimal theta weight for each organelle based on the F1 score. The median theta weight was (0, 0.75, 0.5, 0.5, 1, 1, 0.5, 0.5, 0.5, 0.5, 1, 1) for the cytosol, ER, Golgi, lysosome, mitochondrion, nucleus, chromatin, peroxisome, PM, proteasome, ribosome 40S and ribosome 60S, respectively.

**QSep analysis.** The QSep function which is available as part of the pRoloc R package[33] was used to quantify the resolution of the LOPIT-DC and hyperLOPIT datasets. QSep calculates cluster separation by comparing the average Euclidean distances within and between subcellular clusters. These distances always refer to one specific organelle marker cluster and the distances within clusters are usually smaller than the ones between clusters, except in cases of overlapping subcellular niches. To enable reliable comparison of all distances within a single experiment but also across different studies, QSep further divides each value by the reference within-cluster average distance (for more details see ref. [62]). The resulting distance value is informative of how much the average distance between two clusters is greater than the average distance within a cluster, the reference within-cluster distance here being a measure of how compact a cluster is. The resolution metric used by QSep is not influenced by the number of classes used for its computation and performs consistently well when provided with different organelle marker annotation. However, subcellular marker definition does affect the resolution assessment scoring, with low quality marker lists yielding suboptimal results[62].

**GO term enrichment analysis.** GO over-representation analysis for the unclassified proteins was conducted using the R package goseq[76]. This package was originally developed to account for the relationship between the probability of a differentially expressed gene in RNA-seq experiments and the length of the gene by calculating a probability weight function to estimate the relationship between gene

length and P(differential expression) and then approximating a null distribution for the number of genes expected to be differentially expressed from a given set (e.g. GO term) based on their length alone. An empirical p-value is derived by comparing the number of observed genes to the null expectation. The package allows this approach to be generalised to any observation and any confounding factor. We used protein abundance (derived from[77] taking the maximum abundance recorded across the replicates) and confirmed that more abundant proteins are more likely to be detected in the LOPIT experiments. Proteins not present in the above reference dataset were excluded from the analysis. Resultant p-values were adjusted to account for multiple testing using the Benjamini-Hochberg FDR procedure[78]. GO-terms with adjusted p-value <0.01 and at least five proteins were considered significantly over-represented. Over-representation values shown in figures are not adjusted for protein abundance.

**Plotting of biological feature subcellular localisation.** All signalling and metabolic pathways presented in this manuscript were plotted according to information available in the KEGG PATHWAY database[79].

**Code availability.** No custom code was used to generate, test or process the data described herein. Peptide spectrum matching and quantification to protein-level abundances were performed in Proteome Discoverer v1.4 (Thermo Fisher Scientific) using the Mascot server v2.3.02 (Matrix Science) as described in the Raw data processing and quantification section. Protein localisation analyses were performed using the freely and openly available R Bioconductor packages MSnbase v2.6.1[75] and pRoloc v1.21.9[33] as described in the SVM-based prediction of protein localisation, Data integration by transfer learning and QSep analysis sections in the Methods. Step-by-step tutorials which describe each data analysis stage in detail are also documented with each R package as part of the open-source, open-development Bioconductor project[74] and additional coding examples are available in the Data Analysis section of ref. [63].

**Reporting summary.** Further information on experimental design is available in the Nature Research Reporting Summary linked to this article.

## Data availability

All protein-level datasets generated during this study are available in the R Bioconductor pRolocdata package (version >= 1.19.4) and can be viewed interactively via the pRolocGUI application (version 1.17.0) or directly online through the dedicated R Shiny apps at https://proteome.shinyapps.io/lopitdc-u2os2018 and https://proteome.shinyapps.io/hyperlopit-u2os2018. All proteomics data have been deposited to the ProteomeXchange Consortium via the PRIDE[80] partner repository with the dataset identifier PXD011254 (https://www.ebi.ac.uk/pride/archive/projects/PXD011254). A reporting summary for this Article is available as a Supplementary Information file. All other data supporting the findings of this study are available from the corresponding authors on reasonable request.

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

## Acknowledgements

A.G. was funded through the Alexander S. Onassis Public Benefit Foundation, the Foundation for Education and European Culture (IPEP), the A. G. Leventis Foundation and the Embiricos Trust Scholarship of Jesus College Cambridge. L.M.B. and C.M.M. were supported by a Wellcome Trust Technology Development Grant (grant no. 108467/Z/15/Z). O.M.C. is a Wellcome Trust Mathematical Genomics and Medicine student supported financially by the School of Clinical Medicine, University of Cambridge. O.L. V. is a BBSRC CASE student (BB/R505365/1). K.S.L. is a Wellcome Trust Joint Investigator (grant no. 110170/Z/15/Z). This grant supported L.G. and T.S.S. We would like to thank Mike Deery for performing the mass spectrometry for this project.

## Author contributions

A.G., N.K.B. and K.S.L. designed, planned and performed the experiments and data analyses. T.S.S, L.M.B. and L.G. performed data analyses. A.G., N.K.B. and K.S.L. wrote the manuscript with contributions from T.S.S and L.M.B. Figures were prepared by N.K.B., A.G., T.S.S., L.M.B., O.L.V. and L.G. C.M.M., O.L.V. and O.M.C. advised on the content and layout of the manuscript.

## Additional information

**Competing interests:** The authors declare no competing interests.

