## [Peer Review File · Nature Communications]

Reviewers' comments:

Reviewer #1 (Remarks to the Author):

The manuscript submitted by Geladaki et al. describes a method for subcellular proteome mapping based on differential centrifugation-based fractionation called LOPIT-DC. This method combines the generation of fractions from a cell lysate using different centrifugations, with TMT-labelling for quantification. This method is a derivation of their previous method called hyperLOPIT, which is instead based on density gradients. The main advantage of the new method is the reduction in preparation time, as well as the reduction in samples to be analyzed by mass spectrometry and a lower amount of starting material. This is at the cost of overall resolution, although they demonstrate that because of the single mix and mass spectrometry analysis, they are able to quantify more proteins, and still achieve some degree of separation for some of the organelles, such as the nuclear/chromatin proteins.

They apply both the hyperLOPIT and LOPIT-DC to U2OS cells in order to compare both methods in the same laboratory settings using the same instruments for analysis. The data appears to be very consistent and the overlap in identification and characterization of protein complexes is solid. This demonstrates that LOPIT-DC is an interesting alternative to hyperLOPIT, and the authors provide sufficient details for anyone wishing to use this approach. The procedure is well written and easy to follow and should allow researchers to be able to implement this method.

Overall, my concern is the level of novelty provided by the manuscript, as this method provides an incremental improvement over previous reports from the same group. This manuscript might be more suitable for a more specialized or methods journal.

Reviewer #2 (Remarks to the Author):

This manuscript describes a new version of the LOPIT method that the Lilley group has spent so many years developing. LOPIT is, by a very large margin, the field-standard for organellar proteome

mapping. The version described here uses differential centrifugation and shows that the data quality is very similar to their most advanced method, hyperLOPIT, but much faster and simpler to perform. This is a significant advance in the field.

Concerns

1. FDR. The FDR thresholding seems problematic since it relies on data that can be circular - some of the data in UniProt and GO is based on previous proteomics studies, including LOPIT work by the applicants themselves. This can't be good as it would lead to an underestimation of the FDR rate.

2. Artefacts in TMT data. While the SPS MS3 method is an improvement over MS2 for analyzing isobaric tags, there is still a disturbing amount of signal interference. This is likely not ratio compression but rather spurious signals in the (supposedly silent) reporter channels. From what I was able to tell, the TMT labelling strategy in the three biological replicates done here was the same in each replicate. What happens if the order of labelling is changed? How much of an effect does this have on localizations?

3. Large protein complexes. The claim that this approach can resolve large protein complexes is a little hard to buy. Ribosomes will pellet at 100,000+ r.c.f. but even then not quantitatively. Certainly though, the 60S and 40S do seem to be well-separated in Fig. 7 (wasn't totally sure because the colours didn't show up well for me). Could this be due, however, to partitioning of the ribosome between soluble and ER-bound forms? Because it would be the same proteins, the DC profiles of these would appear to be a hybrid of standard ER and cytosolic markers.

The other complexes are harder to believe. They appear to just be in the cytosol. This argument is not very strong.

Reviewer #3 (Remarks to the Author):

This study provides a direct and detailed comparison of two previously published approaches for spatial mapping of the cellular proteome. The first approach, abbreviated hyperLOPIT, is based on the distribution of proteins across multiple density gradient fractions and has been developed and refined by the author group. The hyperLOPIT dataset was published last year in *Science* and supplemented here with a third replicate. The second approach, abbreviated LOPIT-DC, is based on differential centrifugation resulting in multiple pellet fractions. This approach was demonstrated recently by Itzhak et. al. (2016, 2017) and here adapted for 10 plex TMT labelling. The hyperLOPIT and LOPIT-DC methods have been applied to spatial mapping of the human U-2 OS cancer cell line. The

experiments are technically well-performed and the comparative analyses are well-presented using advanced statistical and spatial proteomics data analyses methods developed by the author group and made available as open-source and open-development tools. The overall conclusions from the analysis are in agreement with the studies of Itzhak et. 2016, 2017 that differential centrifugation combined with quantitative proteomics analysis provides a simple, generic, sensitive, and reproducible strategy to generate a spatial map of the subcellular proteome at relative high resolution. The detailed comparison of the hyperLOPIT and LOPIT-DC methods provides further insights into the strength, weaknesses, and synergies of the two strategies, which is convincingly demonstrated in figures based on protein localisation classification and data visualisation (e.g. Figure 4). The study also serves to guide future experiments to explore the spatio-temporal dynamics of cellular proteins.

Minor comments and suggestions:

Table 3 is mentioned before Table 1. Figures S4 is mentioned before Figure S3.

Page 6 “LC-SPS-MS3 analysis of the U-2 OS LOPIT-DC fractions resulted in identification of 9386 protein groups after replicate merging and, following initial processing and missing value removal, 6837 protein groups with a full reporter ion series across all fractions and replicates remained”.

Comment: The LC-SPS-MS3 analysis method should be explained or referenced when first introduced. It would also be useful with a brief description of the method within the experimental sections rather than referring to previous work. Comment: In principle, differential centrifugation should produce fractions that are highly enriched or depleted for specific organelles. It therefore appears critical to exclude data not represented with a full TMT reporter ion series. This also concerns the arguments presented for TMT labelling: Does mixing of all samples compromise the detection of proteins from low copy number organelles enriched in certain fractions when analysed together with highly abundant proteins present in other fractions? These comments could be discussed in the manuscript.

Figure 2a. “Principal components 1 and 2 separate the organelle marker proteins into three groups”.

Comment: It would be useful with a short description of how the PC dimensions are derived from the data and in Figure 2a it would be useful with an indication of the three groups referred to in the text when 10 annotations are shown NeXT to the figure.

Figure 3a, b. For an unbiased comparison, the two figures should ideally contain the same parameter groups (organelles) although LOPIT-DC for various reasons is less tailored to nuclear sub-structures. For Figure 3c the scale on the two axes should be the same.

Replace LOPITs with LOPIT methods

Proteins involved in signalling pathway plotted upon the LOPIT-DC and hyperLOPIT datasets with assigned proteins are interesting. For Figure 7c and S9-S15, the information content and comparison between the two methods could be increased if the individual signalling proteins were numbered. The observation that subunits of large protein complexes and proteins of signalling pathways are situated close to the same subcellular niche in both datasets suggest subcellular compartmentalization or organelle association. Considering that the manuscript mostly focus on a method comparison, it would be interesting if these observations were further evaluated and discussed to illustrate the value of the spatial datasets as a resource.

We thank the reviewers for their positive comments about our manuscript and have produced a point by point response to their comments as follows:

Reviewers' comments:

Reviewer #1 (Remarks to the Author):

The manuscript submitted by Geladaki et al. describes a method for subcellular proteome mapping based on differential centrifugation-based fractionation called LOPIT-DC. This method combines the generation of fractions from a cell lysate using different centrifugations, with TMT-labelling for quantification. This method is a derivation of their previous method called hyperLOPIT, which is instead based on density gradients. The main advantage of the new method is the reduction in preparation time, as well as the reduction in samples to be analyzed by mass spectrometry and a lower amount of starting material. This is at the cost of overall resolution, although they demonstrate that because of the single mix and mass spectrometry analysis, they are able to quantify more proteins, and still achieve some degree of separation for some of the organelles, such as the nuclear/chromatin proteins. They apply both the hyperLOPIT and LOPIT-DC to U2OS cells in order to compare both methods in the same laboratory settings using the same instruments for analysis. The data appears to be very consistent and the overlap in identification and characterization of protein complexes is solid. This demonstrates that LOPIT-DC is an interesting alternative to hyperLOPIT, and the authors provide sufficient details for anyone wishing to use this approach. The procedure is well written and easy to follow and should allow researchers to be able to implement this method.

Thank you very much for your positive comments.

Overall, my concern is the level of novelty provided by the manuscript, as this method provides an incremental improvement over previous reports from the same group. This manuscript might be more suitable for a more specialized or methods journal.

We view the LOPIT-DC technology as very different from hyperLOPIT. It brings sound experimental design to a wide audience and will be highly enabling where resources, both in terms of sample size, equipment and time are limiting. This is especially important for dynamic studies, where using hyperLOPIT is harder as this method requires considerably more time to execute meaning that some rapid protein relocalisation events may be lost. Furthermore, hyperLOPIT requires more starting material, which is often not available in large enough quantities when working with primary or patient-derived tissues. We therefore do not consider LOPIT-DC to be just an incremental change to the more established hyperLOPIT workflow. We thus believe that the new LOPIT-DC method will benefit a wider audience as it will enable easier and quicker studies of the spatial of proteome. In addition to the experimental protocol associated with LOPIT-DC, we also present a streamlined informatics pipeline, elements of which, such as transfer learning, will be of interest to a broad readership.

Reviewer #2 (Remarks to the Author):

This manuscript describes a new version of the LOPIT method that the Lilley group has spent so many years developing. LOPIT is, by a very large margin, the field-standard for organellar proteome mapping. The version described here uses differential centrifugation and shows that the data quality is very similar to their most advanced method, hyperLOPIT, but much faster and simpler to perform. This is a significant advance in the field.

We are delighted by this reviewer's support of our work

Concerns:

1. FDR. The FDR thresholding seems problematic since it relies on data that can be circular - some of the data in UniProt and GO is based on previous proteomics studies, including LOPIT work by the applicants themselves. This cannot be good as it would lead to an underestimation of the FDR rate.

We have added a sentence to the methods sections of the manuscript to describe how markers were selected and to reassure the reader that no ‘self-prophecy’ was likely by applying LOPIT derived markers to new LOPIT datasets.

2. Artefacts in TMT data. While the SPS MS3 method is an improvement over MS2 for analyzing isobaric tags, there is still a disturbing amount of signal interference. This is likely not ratio compression but rather spurious signals in the (supposedly silent) reporter channels. From what I was able to tell, the TMT labelling strategy in the three biological replicates done here was the same in each replicate. What happens if the order of labelling is changed? How much of an effect does this have on localizations?

In all datasets collected to date we have used a similar labelling strategy and not repeated labelling the same fraction with tags applied in significantly different order. It is however, probably rare to have silent channels in our LOPIT channels as we do not aim to purify different subcellular niches, but to apportion them such that they have different profiles throughout subcellular fractions, thus we would anticipate some signal for each tag.

Furthermore, although TMT quantification is performed at the peptide level, we aggregate to protein-level abundance estimates. Thus, any signal interferences will likely have little impact on the final protein-level abundance profile and resultant localisation assignment

3. Large protein complexes. The claim that this approach can resolve large protein complexes is a little hard to buy. Ribosomes will pellet at 100,000+ r.c.f. but even then not quantitatively. Certainly though, the 60S and 40S do seem to be well-separated in Fig. 7 (wasn't totally sure because the colours didn't show up well for me).

To view separation of the ribosomes we refer the reviewer to Figure 2b and the protein profiles which are shown for each replicate in supporting Figure S1a.

Could this be due, however, to partitioning of the ribosome between soluble and ER-bound forms? Because it would be the same proteins, the DC profiles of these would appear to be a hybrid of standard ER and cytosolic markers.

We do not make an effort in either protocol to maintain the ribosomes bound to the ER and do not see a “peak” in ribosome abundance overlapping the ER markers or cytosol markers (Figure S1).

The other complexes are harder to believe. They appear to just be in the cytosol. This argument is not very strong.

We disagree with the reviewer and have highlighted additional complexes in the revised version of our manuscript with complexes shown that are classified to subcellular niches other than the cytosol.

To reinforce the fact that co-localising members of the same protein complex can be visualised in the case of complexes from very different subcellular locations we have now included more non-cytosolic complexes in Figure 7b. The observed location is identical for both the LOPIT-DC and hyperLOPIT data.

Moreover, to add support for the ability of both LOPIT workflows to return data on co-localisation of complex components from membrane bound organelles, we have added an additional Figure (Figure S11) not only demonstrating co-localisation in the case of four multi-protein complexes, 1) ATP synthase (mitochondrial) 2) N-oligosaccharyl transferase (OST)(ER), 3) Origin recognition complex (nuclear), 4) snRNPs (nuclear) but also showing supporting orthogonal data for each component from the Cell Atlas immune fluorescence database.

We would also like to note the several examples of protein complexes that despite not being classified as residents of a single location, components still cluster within the non-assigned parts of the plots, for example aminoacyl-tRNA synthetase complex (Figure S10).

We hope that the reworking and additions of the above will set the reviewer's mind at rest.

Reviewer #3 (Remarks to the Author):

This study provides a direct and detailed comparison of two previously published approaches for spatial mapping of the cellular proteome. The first approach, abbreviated hyperLOPIT, is based on the distribution of proteins across multiple density gradient fractions and has been developed and refined by the author group. The hyperLOPIT dataset was published last year in Science and supplemented here with a third replicate. The second approach, abbreviated LOPIT-DC, is based on differential centrifugation resulting in multiple pellet fractions. This approach was demonstrated recently by Itzhak et. 2016, 2017 and here adapted for 10 plex TMT labelling. The hyperLOPIT and LOPIT-DC methods have been applied to spatial mapping of the human U-2 OS cancer cell line. The experiments are technically well-performed and the comparative analyses are well-presented using advanced statistical and spatial proteomics data analyses methods developed by the author group and made available as open-source and open-development tools. The overall conclusions from the analysis are in agreement with the studies of Itzhak et. 2016, 2017 that differential centrifugation combined with quantitative proteomics analysis provides a simple, generic, sensitive, and reproducible strategy to generate a spatial map of the subcellular proteome at relative high resolution. The detailed comparison of the hyperLOPIT and LOPIT-DC methods provides further insights into the strength, weaknesses, and synergies of the two strategies, which is convincingly demonstrated in Figures based on protein localisation classification and data visualisation (e.g. Figure 4). The study also serves to guide future experiments to explore the spatio-temporal dynamics of cellular proteins.

Again we are very pleased to read these very supportive comments from this reviewer

Minor comments and suggestions:

Table 3 is mentioned before Table 1. Figure S4 is mentioned before Figure S3.

We thank the reviewer for their careful reading of the manuscript and have changed Table 3 to Table 1 and vice versa. Likewise we have renamed Figure S4 to S3 and vice versa.

Page 6; "LC-SPS-MS3 analysis of the U-2 OS LOPIT-DC fractions resulted in identification of 9386 protein groups after replicate merging and, following initial processing and missing value removal, 6837 protein groups with a full reporter ion series across all fractions and replicates remained".

Comment: The LC-SPS-MS3 analysis method should be explained or referenced when first introduced. It would also be useful with a brief description of the method within the experimental sections rather than referring to previous work.

We have now added the reference to the revised manuscript and also described the experimental parameters we used in the methods section.

Comment: In principle, differential centrifugation should produce fractions that are highly enriched or depleted for specific organelles. It therefore appears critical to exclude data not represented with a full TMT reporter ion series. This also concerns the arguments presented for TMT labelling: Does mixing of all samples compromise the detection of proteins from low copy number organelles enriched in certain fractions when analysed together with highly abundant proteins present in other fractions? These comments could be discussed in the manuscript.

In both the hyperLOPIT and LOPIT-DC protocols we take 50 µg per fraction, thus very abundant organelles are less represented as a percentage of the overall than the less abundant organelles. This means that proteins from less abundant subcellular niches are less prone to be 'overwhelmed' by the proteins from more abundant subcellular niches. As subcellular

location is based on the correlation profiles of location specific marker proteins, taking variable proportions from each fraction will not impinge on the ability to determine these profiles and classify proteins to the correct location.

Figure 2a. “Principal components 1 and 2 separate the organelle marker proteins into three groups”. Comment: It would be useful with a short description of how the PC dimensions are derived from the data and in Figure 2a it would be useful with an indication of the three groups referred to in the text when 10 annotations are shown next to the Figure.

We have added a short description where principal components analysis is first mentioned to describe briefly to readers how the principal components are generated. The sub-grouping referred to in Figure 2a of subcellular niches is very subjective and yes we indeed we see resolution of 10 subcellular niches in the LOPIT-DC dataset, and 12 niches in the hyperLOPIT dataset. We have added a sentence where each dataset is first described to make this clear.

Figure 3a, b. For an unbiased comparison, the two Figures should ideally contain the same parameter groups (organelles) although LOPIT-DC for various reasons is less tailored to nuclear sub-structures.

Throughout the manuscript we have analysed the two datasets with 10 classes for the LOPIT-DC dataset and 12 classes for the hyperLOPIT dataset, which is explained at the beginning of the section entitled “LOPIT-DC achieves high subcellular resolution” as we lack sub-organelle resolution for the nucleus and ribosome in the LOPIT-DC data. As requested by the reviewer we have also added a new Figure (Figure S6) displaying the QSep distances where we have subsetted the LOPIT-DC data into 12 classes to match the hyperLOPIT data. For Figure 3c the scale on the two axes should be the same.

We have changed Figure 3b (Figure 3c in the original version of the manuscript) as requested.

Replace LOPITs with LOPIT methods.

We have changed the revised manuscript as requested.

Proteins involved in signalling pathway plotted upon the LOPIT-DC and hyperLOPIT datasets with assigned proteins are interesting. For Figure 7c and S9-S15, the information content and comparison between the two methods could be increased **if the individual signalling proteins were numbered**. The observation that subunits of large protein complexes and proteins of signalling pathways are situated close to the same subcellular niche in both datasets suggest subcellular compartmentalization or organelle association. Considering that the manuscript mostly focus on a method comparison, it would be interesting if these observations were further evaluated and discussed to illustrate the value of the spatial datasets as a resource.

We were not entirely certain what the reviewer was suggesting by the above comments, however, as in our response to reviewer 2, we have added to and improved Figures 7b and S11 showing co-localisation of components of multiprotein complexes and validated the co-localisation of four complexes as stated above with data from the Cell Atlas database. In response to the request from reviewer 3 we have also numbered the components of the p53 signalling pathway in Figure 7c.

REVIEWERS' COMMENTS:

Reviewer #2 (Remarks to the Author):

The authors have addressed my comments adequately.

Leonard Foster